# Sublinear Algorithms for Estimating Wasserstein and TV Distances: Applications to Fairness and Privacy Auditing

**Debabrota Basu**                                  *debabrota.basu@inria.fr*
*Équipe Scool, Univ. Lille, Inria,*
*CNRS, Centrale Lille, UMR 9189- CRIStAL*
*F-59000 Lille, France*

**Debarshi Chanda**                                 *debarshi.chanda.1997@gmail.com*
*Indian Statistical Institute*
*Kolkata, India*

Reviewed on OpenReview: *https: // openreview. net/ forum? id= m26nTKlpCr*

## Abstract

Resource-efficiently computing representations of probability distributions and the distances between them while only having access to the samples is a fundamental and useful problem across mathematical sciences. In this paper, we propose a generic framework to learn the probability and cumulative distribution functions (PDFs and CDFs) of a sub-Weibull, i.e. almost any light- or heavy-tailed, distribution while the samples from it arrive in a stream. The idea is to reduce these problems into estimating the frequency of an *appropriately chosen subset* of the support of a *properly discretised distribution*. We leverage this reduction to compute mergeable summaries of distributions from the stream of samples while requiring only sublinear space relative to the number of observed samples. This allows us to estimate Wasserstein and Total Variation (TV) distances between any two distributions while samples arrive in streams and from multiple sources. Our algorithms significantly improves on the existing methods for distance estimation incurring super-linear time and linear space complexities, and further extend the mergeable summaries framework to continuous distributions with possibly infinite support. Our results are tight with respect to the existing lower bounds for bounded discrete distributions. In addition, we leverage our proposed estimators of Wasserstein and TV distances to tightly audit the fairness and privacy of algorithms. We empirically demonstrate the efficiency of proposed algorithms across synthetic and real-world datasets.

## 1 Introduction

Computing distances between probability distributions while having access to samples is a fundamental and practical problem in statistics, computer science, information theory, and many other fields. Among the multitude of distances proposed between distributions, we focus on the Wasserstein and Total Variation distances in this work.

**Wasserstein Distance.** Optimal transport (Villani, 2009) has emerged as a popular and useful tool in machine learning. It involves computing Wasserstein distance between distributions and using it further for learning. The statistical and computational aspects of Wasserstein distances have been extensively studied, refer to Panaretos & Zemel (2018); Peyré & Cuturi (2019); Chewi et al. (2024). This geometrically insightful approach has found applications in text document similarity measurement (Kusner et al., 2015), dataset distances (Alvarez-Melis & Fusi, 2020), domain adaption (Courty et al., 2016), generative adversarial networks (Arjovsky et al., 2017), data selection and valuation (Just et al., 2024; Kang et al., 2024), fair

classification and regression (Jiang et al., 2020; Chzhen et al., 2020) to name a few. Thus, it is a fundamental question to compute Wasserstein distance between two distributions when we only have access to samples from it. But computing Wasserstein distance is often memory and computationally intensive until we know the exact parametric form of the data distribution (precisely, location-scatter families) (Gelbrich, 1990; Alvarez-Esteban et al., 2016; Cuesta-Albertos et al., 1996; Alvarez-Melis & Fusi, 2020). In absence of such parametric assumptions, we need linear space complexity and super-linear time complexity in terms of observed samples to estimate Wasserstein distances between the underlying data distributions (Cuturi, 2013; Panaretos & Zemel, 2018; Chizat et al., 2020; Rakotomamonjy et al., 2024; Chewi et al., 2024). Additionally, in practice, the stream might arrive from multiple sources, as in Federated Learning (Kairouz et al., 2021). This has motivated a recent line of work to estimate Wasserstein distance in federated setting (Rakotomamonjy et al., 2024; Li et al., 2024). However, these methods incur communication cost equal to the number of samples in the stream. These gaps in literature motivate us to ask

> Can we estimate the Wasserstein distance between two continuous distributions with infinite support in sublinear space, time, and communication complexity w.r.t. number of samples $n$?

**Total Variation (TV) Distance.** TV (Devroye & Gyorfi, 1990) is another well-studied distance between two probability distributions. It measures the maximum gap between the probabilities of any event w.r.t. the two distributions. TV also quantifies the minimum probability that $X \neq Y$ among all couplings of $(X, Y)$ sampled from the distributions. TV distance between the output distributions of an algorithm operated on two neighbouring datasets also central to auditing the privacy level of the algorithm (Koskela & Mohammadi, 2024). Additionally, in bandits, the hardness of a problem depends on the TV distance between the reward distributions of the available actions (Azize & Basu, 2022; Azize et al., 2023). Thus, estimating TV distance from empirical samples emerges as a fundamental problem in privacy and machine learning. Estimating TV distance has been studied extensively (Canonne, 2022). There is a long line of work focusing on estimating TV distance between discrete distributions with finite support (Kamath et al., 2015; Han et al., 2015; Feng et al., 2024; Devroye & Reddad, 2019; Bhattacharyya et al., 2023; 2024; 2025). Feigenbaum et al. (2002); Guha et al. (2006); Roy & Vasudev (2023) propose sublinear algorithm to estimate TV distance while having access to sample streams. But the question of estimating TV distance for continuous distributions with infinite supports remains wide open. This motivates the question:

> *Can we estimate the TV distance between two continuous distributions with infinite support in sublinear space, time, and communication complexity w.r.t. number of samples $n$?*

We address these questions for any bounded tail distribution by *reducing them into estimation of a sublinear summary, i.e. the frequency of an appropriately chosen subset of the support of a properly discretised distribution.*

**Mergeable Sublinear Representation of Distributions.** We propose to learn an approximately correct sublinear summary of a distribution from the data stream that can address the previous questions (Section 4.1). Because then we can store, communicate, and compute with only these sublinear and discrete summaries of distributions. In literature, we find that the sublinear summaries of histograms has been extensively studied over decades (Misra & Gries, 1982; Alon et al., 1996; Charikar et al., 2002; Cormode & Muthukrishnan, 2005). Additionally, in the continuous distributed monitoring setup (Cormode, 2013), one might want to estimate TV and Wasserstein distances when data streams arrive from multiple sources. Also, with the growth of federated and distributed learning, the question of *how these sublinear-sized summaries of histograms from multiple streams can be merged efficiently* has gained interest (Agarwal et al., 2013; Berinde et al., 2010; Anderson et al., 2017). In machine learning, we work a lot with continuous data distributions. Thus, to be practically useful, we need to extend these sublinear summarisers to continuous setting and find minimal conditions to yield theoretical guarantees. On the other hand, there is another long line of research to learn histograms from samples of a continuous distribution and control the error in this process (Scott, 1979; Freedman & Diaconis, 1981; Ioannidis, 2003; Diakonikolas et al., 2018). But it is an open question to develop algorithms and analysis to join these two streams of research, i.e.

> *Can we learn sublinear summaries of (possibly continuous) distributions while having a stream of $n$ samples from it?*

**Our contributions** address these questions affirmatively.

1. *Computing Mergeable Sublinear Summary from Multiple Streams.* We propose a generic framework to learn summaries of PDF and CDF of a continuous or discrete (possible with infinite support) distribution using sublinear space (Section 4.2). Given data arriving in a stream (or multiple streams) from an underlying distribution, we propose SPA and SCA $\epsilon$-approximate corresponding PDF and CDF independent of stream length. To our knowledge, *we initiate the study of mergeable sublinear summaries over streams from an infinite set* and *establish theoretical error bounds for streams from an infinite set and any sub-Gaussian or sub-Weibull distribution.*

2. *Sublinearly Estimating Wasserstein distance.* We use SCA to propose SWA to PAC-estimate Wasserstein distance (Section 4.3). We first show that *mergeable* sublinear summaries learned by SCA are universal estimators of the true distribution in Wasserstein distance. In turn, SWA computes a sublinear summary of a distribution that is $\mathcal{O}\left(n^{-1/2}\right)$ close in Wasserstein distance using $\widetilde{\mathcal{O}}\left(\sqrt{n}\right)$ space for $n$ samples[1] SWA operates in the distributed/federated setting with a communication cost of $\widetilde{\mathcal{O}}\left(\sqrt{n}\right)$ per round while preserving the estimation guarantee.

3. *Sublinearly Estimating TV distance.* For TV distance, we leverage SPA and set the parameters properly to propose STVA (Section 4.4). STVA maintains mergeable sublinear summaries of the bucketed versions of the true distributions, which have probably infinite buckets in their supports. We show that if we set the bucket width to $\widetilde{\Theta}(n^{-1/3})$, STVA yields a $\widetilde{\mathcal{O}}\left(n^{-1/3}\right)$ estimate of TV distance between two distributions using $\widetilde{\mathcal{O}}\left(n^{1/3}\right)$ space.

4. *Applying estimators for fairness and privacy auditing.* In Section 5.1, we demonstrate usefulness of our Wasserstein and TV distances estimators for auditing fairness and privacy of machine learning models, respectively. Experimental results demonstrate the accuracy and sublinearity of the proposed estimators for auditing models trained on real-life datasets from fairness and privacy literatures.

## 2 Preliminaries

We discuss the fundamentals of distributional distances and the generic algorithmic template to learn mergeable summaries of histograms, which are essential to this work.

**Notations.** $\mu$, $\mu_n$ and $\mu_{b,n}$ denote the true, empirical, and bucketed empirical measure with bucket width $b$, respectively. $p_\mu : \operatorname{Supp}(\mu) \to [0,1]$ and $Q_\mu : \operatorname{Supp}(\mu) \to [0,1]$ denote the probability density function (PDF) and cumulative density function (CDF) of $\mu$, respectively. $[n]$ refers to $\{1, 2, \ldots, n\}$ for $n \in \mathbb{N}$.

### 2.1 Distances between Probability Distributions

Now, we define the Wasserstein and TV distances between distributions. We start with defining the Wasserstein distance between distributions defined over metric spaces.

**Definition 1** (p-th Wasserstein distance (Villani, 2009))**.** Given two probability measures $\mu, \nu$ over a metric space $\mathcal{X}$, the p-th Wasserstein distance between them is

$$\mathcal{W}_{\mathrm{p}}(\mu, \nu) \triangleq \left( \min_{\pi \in \Pi(\mu, \nu)} \int_{\mathcal{X} \times \mathcal{X}} \|x - y\|^p \, \mathrm{d}\pi(x, y) \right)^{\frac{1}{p}} .$$

$\pi$ is a coupling over $x$ and $y$, and $\Pi$ is the set of all couplings.

Here, Wasserstein distance is the cost calculated for the optimal coupling between the two probability measures over Euclidean distances. For univariate distributions, Wasserstein distance can be expressed as norms between inverse CDFs of the distributions (Peyré & Cuturi, 2019).

**Lemma 2.** *For univariate measures $\mu, \nu$ over $\mathbb{R}$, the p-th Wasserstein distance $\mathcal{W}_{\mathrm{p}}(\mu, \nu)$ is*

$$\mathcal{W}_{\mathrm{p}}(\mu, \nu) = \left\| Q_\mu^{-1} - Q_\nu^{-1} \right\|_{L^p([0,1])}^p , \tag{1}$$

---

[1]Soft-O notations, i.e. $\widetilde{\mathcal{O}}(\cdot)$ and $\widetilde{\Theta}(\cdot)$, ignore the polylogarithmic and other lower order terms.

*where $Q_\mu^{-1} : [0, 1] \to \mathbb{R} \cup \{-\infty\}$ of a probability measure $\mu$ is the pseudoinverse function defined as $Q_\mu^{-1}(r) \triangleq$*
$\min_{x \in \mathbb{R} \cup \{-\infty\}} \{x : Q_\mu(x) \geq r\}$ *for $r \in [0, 1]$.*

The other distance that we study is the Total Variation (TV) distance.

**Definition 3** (Total Variation (TV) Distance). Given measures $\mu, \nu$ with support $\mathcal{X}$, TV distance between them is

$$\mathrm{TV}(\mu, \nu) \triangleq \sup_{A \subseteq \mathcal{X}} |\mu(A) - \nu(A)| = \frac{1}{2} \|\mu - \nu\|_1.$$

## 2.2 Mergable Summaries of Histograms

The principal algorithmic technique that we use is that of summarising a histogram, which is a long-studied problem (Misra & Gries, 1982; Alon et al., 1996; Charikar et al., 2002; Agarwal et al., 2013; Anderson et al., 2017). We refer to Cormode & Hadjieleftheriou (2008) for an overview.

First, we formally introduce the problem. We consider data arriving in a stream $\zeta$ of length $n$, where each element belongs to a universe $\mathcal{U}$. The $j$-th element of $\mathcal{U}$, i.e. $\mathcal{U}_j$, appears $f_j$ times in the stream $\zeta$. Our goal is to maintain an approximate count of elements $\hat{f}_j$ such that $\left\| \hat{f}_j - f_j \right\|_p$ is small for some $p$. If we separately keep count of all the elements , the problem is trivial and we get $\hat{f}_j = f_j, \forall \mathcal{U}_j \in \mathcal{U}$. However, the goal is to obtain a 'good' approximation while using sublinear, i.e. $o(n)$, space. More generally, the $i$-th element of the stream can have a weight $\boldsymbol{w}_i \in \mathbb{N}$. Then, the task is to generate estimates $\widehat{\mathcal{F}} = \left\{ \hat{f}_j | \mathcal{U}_j \in \mathcal{U} \right\}$, which are close to the true frequency $f_j = \sum_{i=1}^n \boldsymbol{w}_i \mathbf{1}[\zeta_i = \mathcal{U}_j]$ for all $j$.

Now, we extend the problem setup to aggregate from multiple streams (Agarwal et al., 2013). Given $S$-streams of data $\zeta_1, \zeta_2, ..., \zeta_S$, we aim to generate estimates $\widehat{\mathcal{F}}_1, \widehat{\mathcal{F}}_2, ..., \widehat{\mathcal{F}}_S$ and *combine them efficiently* to output a globally 'good' frequency estimate $\widehat{\mathcal{F}} = \mathtt{merge}\left(\widehat{\mathcal{F}}_1, \widehat{\mathcal{F}}_2, ..., \widehat{\mathcal{F}}_S\right)$ for the concatenated stream $\zeta_1 \circ \zeta_2 \circ ... \circ \zeta_S$.

We build on the algorithm of Anderson et al. (2017), which extends the well-known Misra-Gries algorithm (Misra & Gries, 1982). We refer to it as the Mergeable Misra-Gries (`MMG`) algorithm and provide the details in Algorithm 5. This family of algorithms is broadly known as the *counter-based algorithms for histogram summarisation* and consists of *three main modules*. First, they operate by maintaining a sublinear number of counters, say $k = o(n)$, all initiated with 0. Then, the module `MMG · Update` updates them according to the elements in stream. Specifically, for each stream element, if our algorithm has already assigned a counter, it adds the weight to the corresponding counter. If not, it assigns an empty counter to the element if there is an unassigned counter. Otherwise, the values of all counters are reduced by the median value of counters. Second, if the streams are coming from multiple sources, the module `MMG · Merge` takes the counters in all the summaries as an input stream, and update each counter in the final summary by considering the counters as stream elements and their counts as the corresponding weights. For updating, it reuses the module `MMG · Update`. Finally, given an element, the module `MMG · Estimate` returns an estimate of its frequency by returning the value stored in the counter if the element is assigned a counter, and 0 otherwise. Note that `Update`, `Estimate`, and `Merge` takes amortised $\widetilde{\mathcal{O}}(1), \widetilde{\mathcal{O}}(1)$, and $\widetilde{\mathcal{O}}(k)$ time to execute, respectively. For brevity, we defer the further details to Appendix A.

Now, we derive a rectified version of Theorem 2 of Anderson et al. (2017) that bounds the error in frequency estimation using the residuals $\mathrm{res}(n, k/4)$ (defined below).

**Lemma 4** (Estimation Guarantee of `MMG`). *(a) If `MMG` uses $k$ counters, `MMG · Estimate` yields a summary $\{\hat{f}_j\}_{\mathcal{U}_j \in \mathcal{U}}$ satisfying $0 \leq f_j - \hat{f}_j \leq \frac{\mathrm{res}(n, \tau)}{(k - k^*) - \tau}$, for all $\tau \leq (k - k^*)$ and $\mathcal{U}_j \in |\mathcal{U}|$. Here, $\mathrm{res}(n, \tau)$ denotes the sum of frequency of all but $\tau$ most frequent items. (b) If we choose $k^* = \frac{k}{2}$ $\tau = k/4$, we get that $0 \leq f_j - \hat{f}_j \leq \frac{4 \ \mathrm{res}(n, k/4)}{k}, \forall \mathcal{U}_j \in |\mathcal{U}|$.*

We *extend and analyse `MMG`'s design technique, originally developed for the discrete distributions with finite support points, to continuous and discrete distributions with infinite support points.*

# 3 Problem: Estimating Distances between Distributions from Sample Streams

Now, we formally state our problem setup. Let $\mu$ and $\nu$ be two measures on $\mathbb{R}$. We consider that the data is arriving in two streams $\zeta_\mu$ and $\zeta_\nu$ of size $n$, and each element of the two streams are independent and identically distributed (i.i.d.) samples of $\mu$ and $\nu$, respectively. We denote by $\hat{\mu}_n$ and $\hat{\nu}_n$ the two summaries of $\mu$ and $\nu$ computed from the sample streams. Given a probability metric $\mathfrak{D}$, our objective is to compute an $(\varepsilon, \delta)$-estimate of $\mathfrak{D}(\mu, \nu)$ while using sublinear ($o(n)$) space to store $\hat{\mu}_n$ and $\hat{\nu}_n$. Specifically, we want to yield $\mathfrak{D}(\hat{\mu}_n, \hat{\nu}_n)$ ensuring

$$\mathbb{P}\left[\left|\mathfrak{D}(\mu, \nu) - \mathfrak{D}(\hat{\mu}_n, \hat{\nu}_n)\right| \geq \varepsilon\right] \leq \delta, \tag{2}$$

such that $|\mathrm{essSup}(\hat{\mu}_n) \cup \mathrm{essSup}(\hat{\nu}_n)| = o(n)$, and $(\varepsilon, \delta) \in (0, 1) \times (0, 1)$. In this work, we particularly focus on two probability metrics: Wasserstein distances $\mathfrak{D}(\mu, \nu) = \mathcal{W}_p(\mu, \nu)$, and TV distance $\mathfrak{D}(\mu, \nu) = \mathrm{TV}(\mu, \nu)$. For streams coming from multiple sources, like in the federated (or continuous distributed monitoring) setting, we consider the concatenated streams $\zeta_\mu = \zeta_{\mu,1} \circ \zeta_{\mu,2} \circ \ldots \circ \zeta_{\mu,S}$ and $\zeta_\nu = \zeta_{\nu,1} \circ \zeta_{\nu,2} \circ \ldots \circ \zeta_{\nu,S}$, to be the input streams of length $n$ each to estimate the distance $\mathfrak{D}(\mu, \nu)$.

**Structural Assumptions.** In this work, we assume that the data generating distributions have bounded tails (Assumption 7). For the case of $\mathfrak{D}(\mu, \nu) = \mathcal{W}_p(\mu, \nu)$, we assume the distribution to have $\ell_\mathcal{D}$ bi-Lipschitz CDF (Assumption 9). For the case of $\mathfrak{D}(\mu, \nu) = \mathrm{TV}(\mu, \nu)$, we assume the distribution to have $\ell_\mathcal{D}$-Lipschitz PDF (Assumption 10).

To formalise the bounded-tail assumption, we first define the sub-Gaussian and sub-Weibull random variables.

**Definition 5** ($(\tau, \sigma)$-Sub-Gaussian Distributions). A distribution $\mu$ is said to be $(\tau, \sigma)$-sub-Gaussian if for any $\tau \geq t \geq 0$, we have $\mathbb{P}_{X \sim \mathcal{D}}\left[|X - \mathbb{E}[X]| \geq t\right] \leq 2 \exp\left(-\frac{t^2}{\sigma_\mu^2}\right)$.

When $\tau \to \infty$, $(\tau, \sigma)$-sub-Gaussian distribution reduces to the classical definition of sub-Gaussian distribution (Vershynin, 2018). Sub-Gaussians cover a wide-range of distributions including any bounded distribution, Gaussians, mixture of Gaussians etc. (Vershynin, 2018). It is standard to assume the noise and data to be sub-Gaussian in regression problems (Wainwright, 2019). Sub-Gaussianity appears in the underlying scoring mechanism for classification (Wang et al., 2018). We also consider the notion of sub-Weibull distribution that generalises the notion of sub-Gaussianity to any form of exponentially bounded tails. While sub-Gaussian distributions are considered to be light-tailed, sub-Weibull distributions are considered for heavy-tail distributions in the concentration of measure literature (Foss et al., 2011; Vladimirova et al., 2020; Bakhshizadeh et al., 2023).

**Definition 6** ($(\tau, \alpha)$-Sub-Weibull Distributions). A distribution $\mu$ is said to be $(\tau, \alpha)$-sub-Weibull if there exists some constant $c_\alpha$ for any $\tau \geq t \geq 0$, we have $\mathrm{Pr}_{X \sim \mu}[X \geq t] \leq c_\alpha \exp\left(-t^{1/\alpha}\right)$ ..

When $\tau \to \infty$, $(\tau, \alpha)$-sub-Weibull distribution reduces to the classical definition of sub-Weibull distribution (Vladimirova et al., 2020). For simplicity, we denote $(\infty, \alpha)$-sub-Weibull Distributions as $\alpha$-sub-Weibull. Note that, for sub-Gaussians, $\alpha = 1/2$. Now, we formally state our assumptions.

**Assumption 7** (Sub-Gaussian or sub-Weibull Data Generating Distributions). *We consider the distributions yielding the samples are either $\sigma$-sub-Gaussian or $\alpha$-sub-Weibull.*

The tail bound assumption is required because we have access to only the samples from the stream rather than the true distribution. Thus, we need to have *enough* samples such that the empirical measures available to the algorithm are *close* to the true distributions. The tail bounds provide a control of this concentration of measure phenomenon. We now state our assumptions regarding the lipschitzness of the distributions.

**Definition 8** ($\ell$-Lipschitz Functions). Given two metric spaces $(\mathbf{X}, d_x)$ and $(\mathbf{Y}, d_y)$, a function $f : \mathbf{X} \to \mathbf{Y}$ is said to be $\ell$-Lipschitz if $d_y(f(x_1), f(x_2)) \leq \ell d_x(x_1, x_2) \ \forall x_1, x_2 \in \mathbf{X}$.

**Assumption 9** (Bi-Lipschitz Distributions). *The distribution $\mathcal{D}$ over $(\mathbf{X}, d_x)$ with CDF $Q_\mathcal{D}$ has $\ell_\mathcal{D}$-bi-Lipschitz CDF, if $\frac{1}{\ell_\mathcal{D}} d_x(a, b) \leq |Q_\mathcal{D}(a) - Q_\mathcal{D}(b)| \leq \ell_\mathcal{D} d_x(a, b), \ \forall a, b \in \mathbf{X}$.*

This essentially implies that the PDF of $\mathcal{D}$ is bounded above and below everywhere. Any distribution with bounded PDF, and continuous and compact support satisfies this, which encompasses most of the common data distributions.

**Assumption 10** (Lipschitz PDF Distributions)**.** *The distribution $\mathcal{D}$ with pdf $\mu_{\mathcal{D}}$ over $(\mathbf{X}, d_x)$ has $\ell_{\mathcal{D}}$-Lipschitz PDF, i.e. $|\mu_{\mathcal{D}}(a) - \mu_{\mathcal{D}}(b)| \leq \ell_{\mathcal{D}} d_x(a, b)$ , $\forall a, b \in \mathbf{X}$.*

Any exponential family distribution with bounded parameters have Lipschitz PDF. It includes Gaussians with bounded mean and variance, exponential and Poisson distributions, and other commonly used distributions.

## 4 Sublinear Estimators of Wasserstein and TV Distances

In this section, we explain our methodology for mergeable summary creation of bounded-tail distributions, and estimation of Wasserstein and TV distances from sample streams. We theoretically and numerically demonstrate accuracy and sublinearity of our estimators.

### 4.1 From Sublinear Distance Estimation to Learning Sublinear Summaries

Let us consider the empirical measure $\mu_n$ on $n$ points generated from the true (possibly continuous) measure $\mu$. Thus, for any probability distance $\mathfrak{D}$, we observe that

$$|\mathfrak{D}(\mu, \nu) - \mathfrak{D}(\hat{\mu}_n, \hat{\nu}_n)| \leq \underbrace{\mathfrak{D}(\mu, \mu_n) + \mathfrak{D}(\nu, \nu_n)}_{\text{Concentration of measures}} + \underbrace{\mathfrak{D}(\mu_n, \hat{\mu}_n) + \mathfrak{D}(\nu_n, \hat{\nu}_n)}_{\text{Sublinear summaries}},$$

if all of these distances are well-defined. In that case, the distance of the empirical measure and the true one decreases due to concentration of measures. While we learn sublinear summaries for each of the empirical distributions and aim to control the distance between the empirical distributions and the sublinear summary, i.e. $\mathfrak{D}(\mu_n, \hat{\mu}_n)$, for any bounded-tail true distribution $\mu$.

In Section 4.2, we propose algorithms to estimate PDFs and CDFs of bounded-tail distributions. This allows us to control $\mathfrak{D}(\mu_n, \hat{\mu}_n)$ and in turn, compute the desired distances if $\mu_n$ has bounded-tails. We first proof that the empirical distribution $\mu_n$ corresponding to a sub-Gaussian or sub-Weibull $\mu$ is also sub-Gaussian or sub-Weibull, respectively.

**Theorem 11** (Empirical Measure is Sub-Gaussian)**.** *Given an $\sigma$-sub-Gaussian true measure $\mu$, the empirical measure $\mu_n$ generated by $X_1, X_2, ..., X_n$ drawn i.i.d. from $\mu$ is $(\tau, \sigma)$-sub-Gaussian with probability at least $1 - \delta$ given $n \geq c \exp\left(\frac{\tau^2}{\sigma^2}\right) \log \frac{1}{\delta}$.*

**Theorem 12** (Empirical Distribution is sub-Weibull)**.** *Given an $\alpha$-sub-Weibull true measure $\mu$, the empirical measure $\mu_n$ generated by $X_1, X_2, ..., X_n$ drawn i.i.d. from $\mu$ is $(\tau, \alpha)$-sub-Weibull with probability at least $1 - \delta$ given $n \geq \frac{\exp\left(\sqrt[\alpha]{\tau}\right)}{12} \log \frac{1}{\delta}$.*

We show that with high probability, the empirical distribution retains sub-Gaussianity and sub-Weibull property of the true measure $\mu$. This result is important for learning sublinear summaries of true distribution from sample streams. The proof is in Appendix B.

### 4.2 Learning Mergable Sublinear Summaries

In this section, we state our results regarding sublinear summary approximation of sub-Gaussian and sub-Weibull distributions satisfying Assumption 7. We use `MMG` (Algorithm 5) that approximates a data stream generated from bounded number of elements. We design appropriate bucketing techniques, and leverage the tail bounds to approximate the PDF and CDF of continuous distributions using this algorithm.

**A. Learning PDF.** We first turn the support of the empirical distribution into a collection of buckets of width $b > 0$, and in turn, construct an empirical bucketed distribution $\mu_{b,n}$ out of the empirical distribution $\mu_n$.

**Definition 13** (**Bucketed Empirical Distribution**). Let $\mathcal{D}$ be a distribution supported on a (finite or infinite) closed interval $\mathcal{B} \subseteq \mathbb{R}$ with (discrete or continuous) measure $\mu$. Given a reference point $x_0 \in \mathcal{B}$, bucket width $b$, and an index set $I \subseteq \mathbb{Z}$, we can represent $\mathcal{B} = \mathcal{B}(x_0, I, b) \triangleq \cup_{i \in I}[x_0 + ib, x_0 + (i+1)b]$. We define the bucketed empirical distribution, denoted $\mu_{b,n}$, to be the empirical distribution generated from $n$ samples of $\mu$ as $\mu_{b,n}(i) \triangleq \frac{n_i}{n}$, where $n_i$ is the number of samples falling in the bucket $\sqcup_i$.

For brevity, we denote the bucket $[x_0 + ib, x_0 + (i+1)b)$ by $\sqcup_i$. We note that $\mathcal{B}$ inherits the metric structure of $\mathbb{R}$. Specifically, the distance between two points $x_1 \in \sqcup_i$ and $x_2 \in \sqcup_j$ is defined as $d(x_1, x_2) = |i - j|b$.

In Algorithm 1, we first fix a number of buckets $k$, and treat each of the buckets as an element of the stream. The weight of a bucket is the number of elements in the stream that falls in it, which is $n_i$ for $\sqcup_i$. We apply `MMG` over these buckets to return the estimated frequency $\hat{f}_i$ for each bucket $\sqcup_i$. Finally, we return the learned PDF $\{p_{\mu_{b,n}}(i)\}_{i \in I} = \left\{ \frac{\hat{f}_i}{\max_j \hat{F}_j} \right\}_{i \in I}$ as a summary of $\mu_n$'s PDF. Now, we bound the learning error of `SPA`.

| **Algorithm 1** Sublinear PDF Approximator: | **Algorithm 2** Sublinear CDF Approximator |
|---|---|
| $\text{SPA}(\zeta \overset{i.i.d.}{\leftarrow} \mu, k, b)$ | $\text{SCA}(\zeta \overset{i.i.d.}{\leftarrow} \mu, \varepsilon, k, b)$ |
| 1: **Initialize** | 1: **Initialize** |
| 2: $\text{MMG} \cdot \text{Initialize}(k)$ | 2: $\text{MMG} \cdot \text{Initialize}(k)$ |
| 3: **Update**$(\zeta_i)$ | 3: **Update**$(\zeta_i)$ |
| 4: $\sqcup_{\zeta_i} \leftarrow \sqcup$ that contains $\zeta_i$ | 4: $\sqcup_{\zeta_i} \leftarrow \sqcup$ that contains $\zeta_i$ |
| 5: $\text{MMG} \cdot \text{Update}(\zeta_i, 1)$ | 5: $\text{MMG} \cdot \text{Update}(\sqcup_{\zeta_i}, 1)$ |
| 6: **Merge**$(T_2)$ | 6: **Merge**$(T_2)$ |
| 7: $\text{MMG} \cdot \text{Merge}(T_2)$ | 7: $\text{MMG} \cdot \text{Merge}(T_2)$ |
| 8: **Estimate**$(\zeta_i)$ | 8: **Estimate**$(\zeta_i)$ |
| 9: $\hat{f}_i \leftarrow \text{MMG} \cdot \text{Estimate}(\zeta_i)$ | 9: $\hat{F}_j \leftarrow \text{MMG} \cdot \text{EstimateCumulate}(\zeta_i)$ |
| 10: return $p_{\hat{\mu}_n}(i) \leftarrow \frac{\hat{f}_i}{\max_j \hat{F}_j}$ | 10: return $\widehat{Q}_i \leftarrow \frac{\hat{F}_i}{\max_j \hat{F}_j}$ |

**Theorem 14** (`SPA` Learning Error). *(a) If `SPA` uses $k$ buckets and outputs $\hat{\mu}_n$, then for all $i \in I$,*

$$\left| p_{\hat{\mu}_n}(i) - p_{\mu_{b,n}}(i) \right| \leq \max \left\{ \frac{4 \, \text{res}(n, k/4)}{n} \frac{f_i}{n}, \frac{f_i - \hat{f}_i}{n} \right\}.$$

*(b) Further, if $\mu$ corresponding to $\mu_n$ is $\sigma_\mu$-sub-Gaussian, $|\zeta| = n \geq \frac{c}{\varepsilon} \log\left(\frac{1}{\delta}\right)$, and $k \geq \left\lceil \frac{8\sigma_\mu}{b} \sqrt{\log\left(\frac{1}{\varepsilon}\right)} \right\rceil$, then for all $i \in I$, $\mathbb{P}\left( \left| p_{\hat{\mu}_n}(i) - p_{\mu_{b,n}}(i) \right| \geq \varepsilon \right) \leq \delta$.*

*(c) Further, if $\mu$ corresponding to $\mu_n$ is $\alpha$-SubWeibull, $|\zeta| = n \geq \frac{c}{\varepsilon} \log\left(\frac{1}{\delta}\right)$, and $k \geq \left\lceil \frac{c}{b} \left( \log\left(\frac{1}{\varepsilon}\right) \right)^\alpha \right\rceil$, then for all $i \in I$, $\mathbb{P}\left( \left| p_{\hat{\mu}_n}(i) - p_{\mu_{b,n}}(i) \right| \geq \varepsilon \right) \leq \delta$.*

Lemma 4 shows that the error in `MMG` depends on the sum of frequency of elements except the ones with the highest frequencies. `SPA` leverages this observation, the boundedness of the tails of a distribution (Assumption 7), and the guarantee on the preservation of tail bounds for the empirical measure (Theorem 11 and 12). In `SPA`, we set bucket width $b$ as $k \geq \tau_k(\varepsilon, b)$ such that the (possibly) uncovered tails have sufficiently small mass. This yields an $(\epsilon, \delta)$-PAC estimate of the empirical PDF. The proof is in Appendix C.

**B. Learning CDF.** Now, we want to learn a sublinear summary of the CDF of the empirical distribution $\mu_n$. Thus, we start by bucketing its support with $k$ counters. Given the corresponding PDF $p_{\mu_{b,n}}$, we define the CDF of the bucketed empirical distribution as $Q_{\mu_{b,n}}(i) \triangleq \sum_{j \leq i} p_{\mu_{b,n}}(j)$.

In general, if there is a *total order* on the stream universe $\mathcal{U}$, we can define the cumulant function $F : \mathcal{U} \to \mathbb{N}$ as $F_i \triangleq \sum_{j \leq i} f_j$. Total order exists for the real line and the bucket $\mathcal{B}$ defined on it. Lemma 15 establishes the error bound of `MMG` · `EstimateCumulate` for the cumulant $F$. The proof is in Appendix A.

**Lemma 15** (Error Bound of `MMG` · `EstimateCumulate`). *If `MMG` uses $k$ counters, `MMG` · `EstimateCumulate` yields a summary $\{\hat{F}_j\}_{\mathcal{U}_j \in \mathcal{U}}$ satisfying $0 \leq F_i - \hat{F}_i \leq 2\text{res}(n, \frac{k}{4})$.*

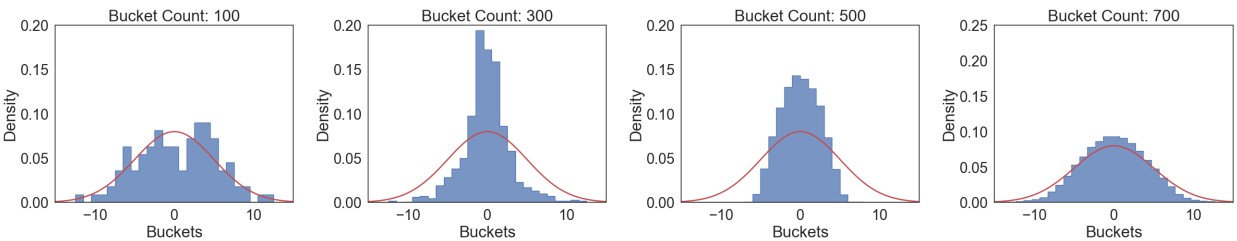

Figure 1: Evolution of summaries constructed using $10^5$ samples from $\mathcal{N}(0,5)$. We set $b = 0.05$ and bucket numbers $100, 300, 500, 700$.

Now, in the spirit of Theorem 14, we appropriately set the bucket width and collect enough samples to yield an $(\epsilon, \delta)$-PAC estimate of the empirical CDF (Appendix C).

**Theorem 16** (SCA Learning Error). *Given stream length $|\zeta| = n \geq \tau_n (\varepsilon, \delta)$ generated from a bounded-tail distribution $\mu$, SCA with bucket width $b$ sets #buckets to $k \geq \tau_k (\varepsilon, b)$ and outputs $\widehat{Q}$, such that with probability at least $1 - \delta$, $\left| Q_{\mu_{b,n}}(i) - \widehat{Q}(i) \right| \leq \varepsilon$ for all $i \in I$.*

| Tail Condition | $\tau_n (\varepsilon, \delta)$ | $\tau_k (\varepsilon, b)$ |
|---|---|---|
| $\sigma_\mu$-sub-Gaussian | $\frac{c}{\varepsilon} \log \left( \frac{1}{\delta} \right)$ | $\left\lceil \frac{8\sigma_\mu}{b} \sqrt{\log \left( \frac{4}{\varepsilon} \right)} \right\rceil$ |
| $\alpha$-sub-Weibull | $\frac{c}{\varepsilon} \log \left( \frac{1}{\delta} \right)$ | $\left\lceil \frac{c}{2b} \left( \log \left( \frac{4}{\varepsilon} \right) \right)^\alpha \right\rceil$ |

Note that the space complexities of both SPA and SCA are independent of streaming universe's size.

**Numerical Demonstration.** We take a stream of $10^5$ samples from a Gaussian $\mathcal{N}(0,5)$ and estimate its PDF with SPA. We set the bucket width to $b = 0.05$ and vary the number of buckets from 100 to 700. Figure 1 validates that increasing the number of buckets, while still being sublinear in #samples, in SPA leads to more accurate summaries of a continuous sub-Gaussian density function.

### 4.3 Estimating Wasserstein Distance

In this section, we introduce an algorithm, namely SWA, to estimate Wasserstein distance between distributions satisfying Assumption 7 and 9. SWA leverages SCA to learn the CDF and use the inverse CDF formulation of Wasserstein (Lemma 2) to obtain a good estimation of the Wasserstein distance between two distributions. We present the pseudocode of SWA in Algorithm 3.

---

**Algorithm 3** Sublinear Wasserstein Estimator: $\text{SWA}(\zeta_\mu, \zeta_\nu \overset{i.i.d.}{\leftarrow} \mu, \nu, \varepsilon, p)$

---

**Require:** $\ell_\mu, \ell_\nu \leftarrow$ Lipschitz constant of $\mu$ and $\nu$, $p \leftarrow p$-th Wasserstein
1: **for** $\mathcal{D} \in \{\mu, \nu\}$ **do**
2: $\quad b_\mathcal{D} \leftarrow \frac{\epsilon}{4}$
3: $\quad \{Q_{\hat{\mathcal{D}}_n}(i)\}_{i=1}^k \leftarrow \text{SCA}(\zeta_\mathcal{D}, b_\mathcal{D}, \tau_k (\varepsilon, \ell_\mathcal{D}, \delta), \varepsilon/2)$
4: **end for**
5: **return** $\mathcal{W}_\text{p} (\hat{\mu}_n, \hat{\nu}_n) \leftarrow$ plug $\{Q_{\hat{\mu}_n}(i)\}_{i=1}^k$ and $\{Q_{\hat{\nu}_n}(i)\}_{i=1}^k$ in Equation (1)

---

**Universal Learning Guarantee of SCA.** We show that SCA is a universally good estimator of the CDF of any bounded-tail measure with respect to the Wasserstein distance. *This is a culmination of three results.*

First, SCA guarantees an $(\epsilon, \delta)$-PAC summary of the CDF of a bucketed empirical distribution. To leverage that guarantee in the case of Wasserstein distance for a (possibly continuous) measure $\mu$, we need to choose the bucket width $b$ appropriately such that $\mathcal{W}_\text{p} (\mu_n, \mu_{b,n})$ is small as the distance of the true and the bucketed distribution is bounded by $b$ (Lemma 32).

Second, once we have the $(\epsilon, \delta)$-PAC summary of the CDF, the rest is to show that it yields small error in computing the inverse CDF, and in turn, the Wasserstein distance (Equation (1)). We establish a guarantee on the approximation error of the pseudoinverse of a bi-Lipschitz CDF, given a guarantee on the approximation error on the CDF (Corollary 34).

Finally, we show that the empirical measure $\mu_n$ corresponding to a true measure $\mu$ with bi-Lipschitz CDF also exhibits bi-Lipschitz CDF with high probability.

**Theorem 17** (Bucketed Empirical Measure is Bi-Lipschitz)**.** *Let the bucketed empirical measure $\mu_{b,n}$ generated by $n$ i.i.d. samples coming from a distribution $\mu$ with $\ell_\mu$ bi-Lipschitz CDF. For any $\varepsilon, \delta \in (0, 1)$ and fixed constant $c > 0$, if $n \geq \frac{c}{\varepsilon^2 b^2} \log\left(\frac{2}{\delta}\right) \max\left\{\frac{1}{\ell_\mu^2}, \ell_\mu^2\right\}$, $\mu_{b,n}$ is $(1 + \varepsilon)\ell_\mu$ bi-Lipschitz with probability $1 - \delta$.*

Proof of Theorem 17 is in Appendix B. Finally, these results together yield Theorem 18, i.e. `SCA` learns a universal estimator of any measure $\mu$. The proof is in Appendix D.

**Theorem 18** (Universal Learning of `SCA` in $\mathcal{W}_{\mathrm{p}}(,)$)**.** *Given $|\zeta| = n \geq \tau_n(\ell_\mu, \varepsilon, \delta)$ generated from a bounded-tail distribution $\mu$ and $\varepsilon, \delta \in (0, 1)$, if we set the bucket width $b = \varepsilon/2$, then `SCA` uses $\tau_k(\ell_\mu, \varepsilon)$ buckets (i.e. space) and outputs $\hat{\mu}_n$, such that $\mathbb{P}(\mathcal{W}_{\mathrm{p}}(\mu_n, \hat{\mu}_n) \geq \varepsilon) \leq \delta$. We omit the constants in $\tau_n(\ell_\mu, \varepsilon, \delta)$ and $\tau_k(\ell_\mu, \varepsilon)$ for simplicity.*

| Tail Condition | $\tau_n(\ell_\mu, \varepsilon, \delta)$ | $\tau_k(\ell_\mu, \varepsilon)$ |
|---|---|---|
| $\sigma_\mu$-sub-Gaussian | $\log(1/\delta) \max\left\{\frac{1}{\varepsilon}, \frac{1}{\ell_\mu^2}, \ell_\mu^2\right\}$ | $\frac{\sigma_\mu}{\varepsilon} \log\left(\frac{\ell_\mu}{\varepsilon}\right)$ |
| $\alpha$-sub-Weibull | $\log(1/\delta) \max\left\{\frac{1}{\varepsilon}, \frac{1}{\ell_\mu^2}, \ell_\mu^2\right\}$ | $\frac{1}{\varepsilon}\left(\log\left(\frac{\ell_\mu}{\varepsilon}\right)\right)^\alpha$ |

**PAC Gurantee of `SWA`.** We know (Equation (3)) that controlling the distance between the empirical measure and the sublinear summary is enough to control the distance estimation error if the empirical distribution concentrates to the true one. Lemma 43 of Bhat & L.A. (2019) proposes the concentration guarantees of empirical measure in Wasserstein distance. Combining this result and Theorem 18 yields estimation guarantees of $\hat{\mu}_n$ w.r.t. true measure $\mu$.

**Theorem 19** (PAC Guarantee of `SWA`)**.** *Given $|\zeta| = n \geq \tau_n(\ell_\mu, \varepsilon, \delta)$ generated from a bounded-tail distribution $\mu$, with bucket width $b = \varepsilon/2$, then `SCA` uses $\tau_k(\ell_\mu, \varepsilon)$ space and outputs $\hat{\mu}_n$ such that $\mathbb{P}(\mathcal{W}_1(\mu, \hat{\mu}_n) \leq \varepsilon) \geq 1 - \delta$. This further shows that for a stream of size $n = \mathcal{O}\left(\epsilon^{-2}\log(1/\delta)\right)$ from two distributions $\mu$ and $\nu$, `SWA` yields*

$$|\mathcal{W}_1(\hat{\mu}_n, \hat{\nu}_n) - \mathcal{W}_1(\mu, \nu)| \leq 4\varepsilon \tag{3}$$

*with probability $1 - 2\delta$. Here, $\epsilon \in (0, 1/4]$ and $\delta \in (0, 1/2]$. Here, $\sigma \triangleq \max\{\sigma_\mu, \sigma_\nu\}$, $\ell \triangleq \max\{\ell_\mu, \ell_\nu\}$. We omit the constants in $\tau_n(\ell_\mu, \varepsilon, \delta)$ and $\tau_k(\ell_\mu, \varepsilon)$ for simplicity.*

| Tail Condition | $\tau_n(\ell_\mu, \varepsilon, \delta)$ | $\tau_k(\ell_\mu, \varepsilon)$ |
|---|---|---|
| $\sigma$-sub-Gaussian | $\log(1/\delta) \max\left\{\frac{1}{\varepsilon^2}, \frac{1}{\ell^2}, \ell^2\right\}$ | $\frac{\sigma}{\varepsilon} \log\left(\frac{\ell}{\varepsilon}\right)$ |
| $\alpha$-sub-Weibull | $\log(1/\delta) \max\left\{\frac{1}{\varepsilon^2}, \left(\log\frac{1}{\delta}\right)^{2\alpha - 1}, \frac{1}{\ell^2}, \ell^2\right\}$ | $\frac{1}{\varepsilon}\left(\log\left(\frac{\ell}{\varepsilon}\right)\right)^\alpha$ |

*Remark* 20 (Space, Time, and Communication Complexity of `SWA`.)**.** `SWA` requires $\widetilde{\mathcal{O}}\left(\frac{1}{\varepsilon}\right)$ time, space, and communication cost to learn a distribution in Wasserstein distance. Observe that the lower bound on $n$ in Theorem 19 ensures convergence $\mu_n$ to $\mu$ in Wasserstein distance. This is the only term that has $\delta$-dependence as the rest of the algorithm is deterministic. `SWA`'s time, space and communication complexities are sublinear $(\widetilde{\Theta}(\sqrt{n}))$ compared to the number of samples required to ensure convergence of the empirical measure while retaining the same error.

## 4.4 Estimating TV Distance

In this section, we introduce `STVA` that yields an estimate of TV distance between two bounded-tail distributions with Lipschitz PDFs. `STVA` leverages `SPA` to sublinearly learn the PDFs of the distributions, and then, estimates the TV distance as $L_1$ norm between them (Definition 3).

---

**Algorithm 4** Sublinear TV Distance Estimator: $\texttt{STVA}(\zeta_\mu, \zeta_\nu \overset{i.i.d.}{\leftarrow} \mu, \nu, \varepsilon, b)$

---

**Require:** $\sigma_\mu, \sigma_\nu \leftarrow$ Subgaussian parameter of $\mu$ and $\nu$, $b \leftarrow$ bucket width
1: **for** $\mathcal{D} \in \{\mu, \nu\}$ **do**
2:    $k_\mathcal{D} \leftarrow \tau_k\left(\varepsilon, \ell, \delta\right)$
3:    $b_\mathcal{D} \leftarrow \tau_b\left(\varepsilon, \ell\right)$
4:    $\{p_{\hat{\mathcal{D}}_n}(i)\}_{i=1}^k = \{\hat{p}_n(i)\}_{i=1}^k \leftarrow \texttt{SPA}\left(\mathcal{D}, k_\mathcal{D}, b_\mathcal{D}\right)$
5: **end for**
6: **return** $\mathrm{TV}\left(\hat{\mu}_n, \hat{\nu}_n\right) \leftarrow 0.5\sum_i \left|p_{\hat{\mu}(i)} - p_{\hat{\nu}(i)}\right| \mathbf{1}\left(\max\left\{p_{\hat{\mu}(i)}, p_{\hat{\nu}(i)} > 0\right\}\right)$

---

**PAC Guarantee of STVA.** *We establish the estimation guarantee of STVA in three steps.* First, we prove that if we choose the bucket width properly to discretise the support of $\mu$, we obtain a control over the TV distance between true distribution and its bucketed version. Second, we leverage the concentration inequalities of Berend & Kontorovich (2013) to control the TV distance between the bucketed distribution and the empirical bucketed distribution from the streamed samples. Finally, we establish the approximation guarantee of SPA to control the TV distance between the empirical bucketed distribution and its sublinear summary.

**Step 1: Setting the Bucket Width.** Standard rules (Scott, 1979; Freedman & Diaconis, 1981) of choosing optimal bucket width of a histogram depends either on rule of thumbs or crucially depends on parameters such as sample size $n$, $\int_{-\infty}^{\infty} p'(x)^2\,dx$ etc., which are difficult to estimate for an unknown distribution arriving in a stream. We propose a bucketing technique depending on the Lipschitz constant of the true PDF and the sub-Gaussian parameter of the true distribution to tune the bucket width a priori. Proof is in Appendix E

**Theorem 21** (Bucket Width for TV Distance Estimation)**.** *Given a bounded-tail distribution $\mu$ with $\ell_\mu$-Lipschitz PDF and a corresponding bucketed measure $\mu_b$, if we fix the bucket width $b = \tau_b\left(\varepsilon, \ell\right)$, we have $\mathrm{TV}\left(\mu, \mu_b\right) \leq \epsilon$.*

| Tail Condition | $\sigma_\mu$-sub-Gaussian | $\alpha$-sub-Weibull |
|---|---|---|
| $\tau_b\left(\varepsilon, \ell_\mu\right)$ | $\frac{\varepsilon}{\sigma_\mu \ell_\mu \sqrt{\log(2/\varepsilon)}}$ | $\frac{\varepsilon}{\ell_\mu(\log(2c_\alpha/\varepsilon))^\alpha}$ |

**Step 2: Concentration of Bucketed Empirical Distribution in TV.** For continuous measures, assumption on the structure of the distribution is necessary to establish meaningful convergence rates in TV distance (Diakonikolas, 2016). We establish a convergence result for bucketed measure $\mu_b$ generated from a bounded-tail measure $\mu$ that possibly has infinite buckets in its support (Appendix E).

**Lemma 22** (Concentration in TV over Infinite Buckets)**.** *Let $\mu_n$ be an empirical measure generated from a discrete bucketed measure $\mu_b$ corresponding to a bounded-tail distribution $\mu$, and, $\varepsilon, \delta \in (0,1)$. Then, for $n \geq \tau_n\left(\varepsilon, b, \delta\right)$, $\mathbb{P}\left[\mathrm{TV}\left(\mu, \mu_n\right) \geq \varepsilon\right] \leq \delta$. Here, $\Gamma(\cdot)$ denotes the Gamma function (Davis, 1959).*

| Tail Condition | $\sigma_\mu$-sub-Gaussian | $\alpha$-sub-Weibull |
|---|---|---|
| $\tau_n\left(\varepsilon, b, \delta\right)$ | $c\varepsilon^{-2} \max\left\{\frac{4\sigma_\mu\sqrt{\pi}}{b}, \log\left(1/\delta\right)\right\}$ | $c\varepsilon^{-2} \max\left\{\frac{2c_\alpha}{b}\Gamma(1+\alpha), \log\left(1/\delta\right)\right\}$ |

**Step 3: Approximation Guarantee of SPA.** Theorem 14 shows that SPA learns a pointwise approximation to the PDF of any bucketed empirical measure. However, to extend this guarantee to the TV distance, we need to show that the the sum of errors over the entire support is small. Note that bounded-tail distributions have small mass in the tails (i.e. in most of the buckets). Hence, it suffices to control the error in learning PDF of the 'heavier' buckets to control the error in estimating the TV distance. The proof is in Appendix E.

**Theorem 23** (Learning Error of SPA)**.** *If SPA accesses a stream of length $|\zeta| = n \geq \tau_n\left(\varepsilon, \delta\right)$ from a bounded-tail distribution, and uses $\tau_k\left(\varepsilon, \delta\right)$ buckets to output a sublinear summary $p_{\hat{\mu}_n}$, then with probability $1 - \delta$, $\mathrm{TV}\left(\hat{\mu}_n, \mu_{b,n}\right) \leq \left\|p_{\hat{\mu}_n} - p_{\mu_{b,n}}\right\|_1 \leq \varepsilon$.*

| Tail Condition | $\tau_n\,(\varepsilon,\delta)$ | $\tau_k\,(\varepsilon,b)$ |
|---|---|---|
| $\sigma_\mu$-sub-Gaussian | $\frac{c}{\varepsilon}\log\left(\frac{1}{\delta}\right)$ | $\left\lceil\frac{8\sigma_\mu}{b}\sqrt{\log\left(\frac{6}{\varepsilon}\right)}\right\rceil$ |
| $\alpha$-sub-Weibull | $\frac{c}{\varepsilon}\log\left(\frac{1}{\delta}\right)$ | $\left\lceil\frac{c}{b}\left(\log\left(\frac{6}{\varepsilon}\right)\right)^\alpha\right\rceil$ |

These results together yield the PAC guarantee for `STVA`.

**Theorem 24** (PAC Guarantee of `STVA`). *Given $\epsilon \in (0, 1/6]$, $\delta \in (0, 1/4]$, a stream of size $n \geq \tau_n\,(\varepsilon, \ell, \delta)$ from two bounded-tail and $\ell$-Lipschitz distributions $\mu$ and $\nu$, and bucket width $b = \tau_b\,(\varepsilon, \ell, \delta)$, `STVA` uses $\tau_k\,(\varepsilon, \ell, \delta)$ space and*

$$\mathbb{P}\left(\left|\,\mathrm{TV}\left(\hat{\mu}_n, \hat{\nu}_n\right) - \mathrm{TV}\left(\mu, \nu\right)\right| \geq 6\varepsilon\right) \leq 4\delta\,. \tag{4}$$

*Here, $\sigma \triangleq \max\{\sigma_\mu, \sigma_\nu\}$, $\ell \triangleq \max\{\ell_\mu, \ell_\nu\}$. We omit the constants in $\tau_n\,(\ell_\mu, \varepsilon, \delta)$ and $\tau_k\,(\ell_\mu, \varepsilon)$ for simplicity.*

| Tail Condition | $\tau_n\,(\varepsilon, \ell, \delta)$ | $\tau_k\,(\varepsilon, \ell)$ | $\tau_b\,(\varepsilon, \ell)$ |
|---|---|---|---|
| $\sigma$-sub-Gaussian | $\varepsilon^{-2}\max\left\{\frac{\sigma^2\ell\log(1/\varepsilon)}{\varepsilon}, \log\left(\frac{1}{\delta}\right)\right\}$ | $\frac{\sigma^2\ell}{\varepsilon}\log\left(\frac{1}{\varepsilon}\right)$ | $\frac{\varepsilon}{\sigma\ell\sqrt{\log(2/\varepsilon)}}$ |
| $\alpha$-sub-Weibull | $\varepsilon^{-2}\max\left\{\frac{\ell(\log(1/\varepsilon))^\alpha}{\varepsilon}\Gamma(1+\alpha), \log\left(\frac{1}{\delta}\right)\right\}$ | $\frac{\ell}{\varepsilon}\left(\log\left(\frac{1}{\varepsilon}\right)\right)^{2\alpha}$ | $\frac{\varepsilon}{\ell(\log(2c_\alpha/\varepsilon))^\alpha}$ |

*Remark* 25 (Space, Time, Communication Complexity of `STVA`). `STVA` requires $\widetilde{\mathcal{O}}\left(\frac{1}{\varepsilon}\right)$ time, space, and communication complexity per round to estimate the TV distance between two bounded-tail distributions[2]. Observe that the lower bound on $n$ ensures convergence of $\hat{\mu}_n$ to $\mu$ in TV distance (Theorem 24). This is the only term with $\delta$-dependence as the rest of the algorithm is deterministic. Thus, `STVA` achieves $\widetilde{\Theta}(n^{1/3})$ time, space, and communication complexity with respect to #samples required for convergence of empirical measure while retaining the order of error.

Note that the bounds in Theorem 24 ensures that $\varepsilon$ is of the order $n^{-1/3}$, which ensures that the bucket width is of the order $n^{-1/3}$ and the error in TV distance is of the order $n^{-1/3}$. Our result is consistent with those of other standard histogram bucket width rules, which chooses bucket width of order $n^{-1/3}$ and ensures the integrated mean squared error to be of order $n^{2/3}$ (Scott, 1979; Freedman & Diaconis, 1981).

*Remark* 26 (Further Implications of Theorem 23). We observe that Theorem 23 does not only yield a guarantee on TV distance estimation error but also have two further implications, which might be of general interest.

1. $\ell_1$-*estimation of True Frequencies:* A closer look into the result shows that `SPA` can also yield an $\ell_1$ approximation of the true frequencies ($\{f_i\}_i$) corresponding to any probability distribution when the tails are sufficiently bounded.

2. *Estimation of any Integral Probability Metric (IPM):* As our algorithms yield pointwise estimates of both the PDF and CDF of a distribution, these methods can be used to estimate any Integral Probability Metric (IPM), such as Dudley metric, Maximum Mean Discrepancy (MMD), Kolmogorov distance (Sriperumbudur et al., 2012).

## 4.5 Tightness of Space Complexity

We compare upper bounds on space complexities of `SPA` and `SCA` to the existing lower bounds for finite universe of streams. Frequency estimation have been extensively studied in this setting. Theorem 27 provides a $\Omega\left(\varepsilon^{-1}\right)$ lower bound on space complexity of any non-trivial solution, i.e. not storing the entire stream, or counts of all elements in the universe, to the frequency estimation problem with respect to the $\ell_\infty$-norm.

---

[2]Note that for sub-Gaussians $\alpha = \frac{1}{2}$.

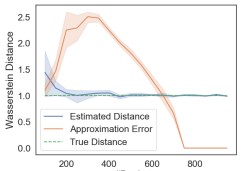 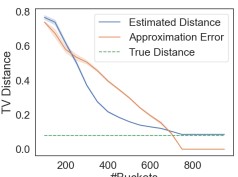 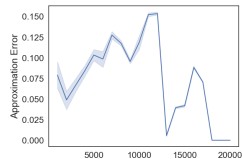 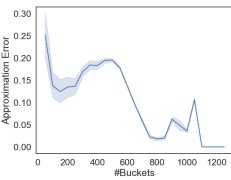 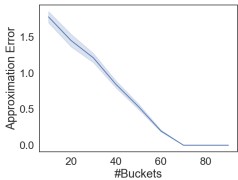

Figure 2: Performance of SWA with $\mathcal{N}(0,5)$, $\mathcal{N}(1,5)$ and $b = 0.05$

Figure 3: Performance of STVA with $\mathcal{N}(0,5)$, $\mathcal{N}(1,5)$ and $b = 0.05$

Figure 4: Auditing with SWA on regression output of ACS_Income .

Figure 5: Auditing with SWA on classification output of ACS_Income .

Figure 6: Privacy Auditing of logistic regression on MNIST.

**Theorem 27** (Lower Bound of Frequency Estimation in $\ell_\infty$-norm (Chakrabarti, 2024)). *Given a stream $\zeta$ of length $n$ coming from a universe $\mathcal{U}$ of size $m$, any algorithm that produces a $(\varepsilon n)$-$\ell_\infty$ approximation of the frequency vector $\{f_i\}_{i \in [m]}$ as $\left\{ \hat{f}_i \right\}_{i \in [m]}$ must use $\Omega \left( \min \left( n, m, \varepsilon^{-1} \right) \right)$ space.*

Our framework provides a $\widetilde{\mathcal{O}} \left( \varepsilon^{-1} \right)$ approximation of the true frequencies in $\ell_1$-norm for any bounded-tail distribution with both infinite and finite supports, i.e. universe of streams (ref. Remark 26)[3]. Hence, Theorem 27 indicates that *our algorithms are tight in terms of space complexity.*

## 5 Experimental Analysis

Our aim is to understand *whether SWA and STVA yield PAC estimates of Wasserstein and TV distances between sub-Gaussian distributions with sublinear space, i.e. number of buckets, with respect to the number of samples.*

**Setup.** We compute distances between two Gaussian distributions $\mathcal{N}(0,5)$ and $\mathcal{N}(1,5)$, where $10^5$ samples from each of them arrive through $S = 10$ sources. True TV and Wasserstein distances between them are 0.0797 and 1.00, respectively. We set the bucket width to $b = 0.05$ and increase the number of buckets as $\{100, 150, \ldots, 1000\}$. We run each of the experiments 50 times. Finally, for SWA, we report the estimated Wasserstein distance and the learning error of SCA, i.e. $\mathcal{W}_p \left( \mu_n, \hat{\mu}_n \right)$, in Figure 2. Similarly, for STVA, we report the estimated TV distance and the learning error of SPA, i.e. $TV \left( \mu_n, \hat{\mu}_n \right)$, in Figure 3. We report the true distances in each case as the dotted horizontal lines.

**Results.**

1. **Accuracy and Sublinearity of SWA.** The estimation error of SWA and learning error becomes negligible as the bucket number reaches to 750 while using $10^5$ samples. This is better than the theoretical upper limit of buckets, i.e. 1040, suggested by our results.

2. **Accuracy and Sublinearity of STVA.** The estimation error of STVA and learning error becomes negligible as #buckets also reaches to 750. Thus, *the results demonstrate that both STVA and SWA yield accurate estimates of TV and Wasserstein distances while using only sublinear space w.r.t. #samples.*

### 5.1 Applications: Auditing Fairness and Privacy

Auditing fairness (Ghosh et al., 2021; 2022; Yan & Zhang, 2022) and privacy (Nasr et al., 2023; Steinke et al., 2024; Koskela & Mohammadi, 2024; Annamalai & Cristofaro, 2024; Azize & Basu, 2025) of Machine Learning (ML) models is an important and increasingly studied question for developing trustworthy ML. Here, we deploy SWA and STVA to estimate the fairness and privacy of ML models trained on real-world data, respectively. The details are provided in Appendix G.

---

[3]Note that while our techniques consider the data arrives from a distribution rather than arbitrarily from a universe (as in Theorem 27), it suffices if the underlying data are generated from a distribution and the order of their arrivals is arbitrary.

# 6    Discussions and Future Works

We propose an algorithmic framework to learn *mergeable* and *sublinear* summaries of discrete and continuous, sub-Gaussian and sub-Weibull distributions while data streams arrive from a single or multiple sources. We show that the computed mergeable sublinear summarisers are universal estimators with respect to Wasserstein distance. We establish the first sublinear time, space, and communication algorithms to estimate TV and Wasserstein distances over continuous and discrete (with infinite support) distributions. We also note that our framework can be used to estimate any $\ell_p$-norm based distance between distributions with sublinear resources. However, our work is constrained to distributions over scalars. A future direction is to study these problems for multi-variate distributions.

## Acknowledgement

Both the authors would like to thank the Inria-ISI Kolkata associate team SeRAI (DRI-012551) for supporting this collaboration. D. Basu acknowledges ANR JCJC project REPUBLIC (ANR-22-CE23-0003-01) and PEPR project FOUNDRY (ANR23-PEIA-0003) partially funding this work. We also thank Uddalok Sarkar and Udvas Das for their support.

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

# Appendix

## Table of Contents

# A    Mergeable Misra Gries Algorithm

---

**Algorithm 5** `MMG` (Anderson et al., 2017)

---

1: **Initialize**(k)
2:   $T \leftarrow \emptyset$                                    ▷ $T$ is the set of items for which counters are assigned
3:   $k \leftarrow k$
4:   $k^* \leftarrow \frac{k}{2}$
5:
6: **Update**$(i, \Delta)$
7: **if** $\zeta_i \in T$ **then**
8:     $c(\zeta_i) \leftarrow c(\zeta_i) + \Delta$
9: **else if** $|T| < k$ **then**
10:     $T = T \cup \{\zeta_i\}$
11:     $c(\zeta_i) \leftarrow c(\zeta_i) + \Delta$
12: **else**
13:     DecrementCounters()
14: **end if**
15: **if** $\Delta \geq c_{k^*}$ **then**
16:     $T = T \cup \{\zeta_i\}$                    ▷ $|T| \leq k^* + 1 \leq k$ after this line
17:     $c(\zeta_i) \leftarrow c(\zeta_i) + \Delta - c_{k^*}$
18: **end if**
19:
20: **DecrementCounters()**
21: // Notation: Let $c_{k^*}$ be the $k^*$-th largest value, counting multiplicity,
22: // in the multiset $\{c(\zeta_j): \zeta_j \in T\}$.
23: **for** $\zeta_j \in T$: **do**
24:     $c(\zeta_j) = c(\zeta_j) - c_{k^*}$
25:     **if** $c(\zeta_j) \leq 0$:
26:     $T = T \setminus \{\zeta_j\}$
27: **end for**
28:
29: **Estimate**$(i)$
30: **if** $\zeta_i \in T$ **then**
31:     **return** $c(\zeta_i)$
32: **else**
33:     **return** 0
34: **end if**
35:
36: **EstimateCumulate**$(\zeta_i)$
37: $c(\zeta_i) \leftarrow$ **Estimate** $(\zeta_i)$
38: **return** $\hat{F}_i \leftarrow \sum_{\zeta_j \leq \zeta_i} C_j$
39:
40: **Merge**$(T_2)$
41: **for** $\zeta_i \in T_2$ **do**
42:     Update$_{T_1}(\zeta_i, c(\zeta_i))$
43:     **return** $T_1$
44: **end for**

---

First, we outline the proof for the Lemma 4, restated here for ease of reading.

**Lemma 4** (Estimation Guarantee of `MMG`)**.** *(a) If* `MMG` *uses $k$ counters,* `MMG · Estimate` *yields a summary* $\{\hat{f}_j\}_{u_j \in \mathcal{U}}$ *satisfying* $0 \leq f_j - \hat{f}_j \leq \frac{\text{res}(n,\tau)}{(k-k^*)-\tau}$, *for all* $\tau \leq (k - k^*)$ *and* $\mathcal{U}_j \in |\mathcal{U}|$. *Here,* $\text{res}(n, \tau)$ *denotes the sum of frequency of all but $\tau$ most frequent items. (b) If we choose $k^* = \frac{k}{2}$ $\tau = k/4$, we get that* $0 \leq f_j - \hat{f}_j \leq \frac{4 \text{ res}(n,k/4)}{k}$, $\forall \mathcal{U}_j \in |\mathcal{U}|$.

The proof of the result follows that of Theorem 2 in Anderson et al. (2017). The change is in the Lemma 4. We provide the proof for the updated statement here. The notation is kept same as that of the original work for ease of reading.

As in the original paper, we define $N_l = \sum_{i=1}^{l} \boldsymbol{w}_j$, $C_l = \sum_{i=1}^{k} c_i$, and $E_l = \sum_{i \in [m]} f_{i,l} - \hat{f}_{i,l}$

**Lemma 28.** $E_n \leq (N_n - C_n) / (k - k^*)$

*Proof.* As in the original proof, we proceed by proof by induction, the case for $l = 0$ is true as $N_o = C_0 = E_0 = 0$.

Suppose the hypothesis holds for some $l - 1$, i.e.:

$$E_{l-1} \leq (N_{l-1} - C_{l-1}) / (k - k^*)$$

If the $l$-th element in the stream does not cause `DecrementCounters` to be called, we have $E_l = E_{l-1}$, $C_l = C_{l-1} + \boldsymbol{w}_l$, $N_l = N_{l-1} + \boldsymbol{w}_l$. Hence ,we have:

$$E_l = E_{l-1} \leq (N_{l-1} - C_{l-1}) / (k - k^*) = (N_l - C_l) / (k - k^*)$$

The case when `DecrementCounters` is called is more interesting, we have

$$E_l = E_{l-1} + c_{k^*} \tag{5}$$

Here, we know that at $l - 1$-th step, $(k - k^*)$ counters had value $\geq c_{k^*}$. Therefore, they retain some non-negative value after being reduced by $c_{k^*}$, while the remaining $k^*$ counters have value 0. Hence, we have $C_l \leq C_{l-1} - (k - k^*) c_{k^*}$.

$$
\begin{aligned}
& N_l - C_l \\
=& N_{l-1} + \boldsymbol{w}_l - C_l \\
\geq& N_{l-1} + \boldsymbol{w}_l - C_{l-1} + (k - k^*) c_{k^*} \\
\geq& N_{l-1} - C_{l-1} + (k - k^*) c_{k^*}
\end{aligned}
\tag{6}
$$

Now, combining Equations (5) and (6), we have:

$$E_l = E_{l-1} + c_{k^*} \leq \frac{N_{l-1} - C_{l-1}}{(k - k^*)} + c_{k^*} = \frac{N_{l-1} - C_{l-1} + (k - k^*) c_{k^*}}{(k - k^*)} \leq \frac{N_l - C_l}{(k - k^*)}$$

$\square$

The second part of the proof of Theorem 2 in the original paper follows similarly under the updated statement of Lemma 28. Giving us the final result of Lemma 4.

To establish the estimation guarantee of `EstimateCumulate`, we need a simple corollary of the Lemma 4:

**Corollary 29** (`MMG` Estimation Guarantees). *For $j \in \{1, 2\}$, given streams $\zeta_j$ being stored in two separate sets of counters of size $k$, the merged summary satisfies:*

$$0 \leq f_i - \hat{f}_i \leq \frac{4 \ \text{res}(n, \frac{k}{4})}{k}$$

*for all $j \leq k^*$, where $\text{res}(n, j)$ denotes the sum frequency of all but $j$ of the most frequent items.*

*Proof.* We fix $k^* = \frac{k}{2}$, and $j = \frac{k}{4}$. These values satisfy the criteria of Theorem 4, i.e. $\frac{k}{2} = \Omega(k)$ and $j < k^*$. Plugging in these values gives the result. $\square$

We now prove the Lemma 15:

*Proof.* We denote by $Top_k$ the set of $k$ elements with highest true frequency $f_i$. Then, we have:

$$F_i - \hat{F}_i = \sum_{j \leq i} f_j - \hat{f}_j$$

$$= \sum_{\substack{j \leq i \\ j \in Top_{\frac{k}{4}}}} f_j - \hat{f}_j + \sum_{\substack{j \leq i \\ j \notin Top_{\frac{k}{4}}}} f_j$$

$$\leq \sum_{j \in Top_{\frac{k}{4}}} \frac{4 \ \text{res}(n, \frac{k}{4})}{k} + \sum_{j \notin Top_{\frac{k}{4}}} f_j$$

$$\leq 2 \ \text{res}(n, \frac{k}{4})$$

Where the first inequality uses Corollary 29, and the last inequality follows from the definition of $\text{res}(n, k)$. $\square$

## B  Properties of Empirical Measure

In this section we provide the proofs for Theorems 11, 12, and 17. We restate the theorems for easy of reading. We introduce the multiplicative Chernoff bound that will be relevant to our analysis.

**Lemma 30** (Multiplicative Chernoff Bound(Mitzenmacher & Upfal, 2005))**.** *Given i.i.d. random variables* $X_1, X_2, ..., X_n$ *where* $\Pr[X_i = 1] \leq p$ *and, define* $X = \frac{1}{n} \sum_{i \in [n]} X_i$. *Then, we have:*

$$\Pr[X \geq (1 + \varepsilon)p] \leq \exp\left(-\frac{n \varepsilon^2 p}{3}\right) \qquad\qquad 0 \leq \varepsilon < 1 \qquad\qquad (7)$$

$$(8)$$

**Theorem 11** (Empirical Measure is Sub-Gaussian)**.** *Given an* $\sigma$-*sub-Gaussian true measure* $\mu$, *the empirical measure* $\mu_n$ *generated by* $X_1, X_2, ..., X_n$ *drawn i.i.d. from* $\mu$ *is* $(\tau, \sigma)$-*sub-Gaussian with probability at least* $1 - \delta$ *given* $n \geq c \exp\left(\frac{\tau^2}{\sigma^2}\right) \log \frac{1}{\delta}$.

*Proof.* To capture the tail behaviour w.r.t. some $t \leq \tau$ for each sample, we define random variables $Z_1, Z_2, ..., Z_n$ as:

$$Z_i = \begin{cases} 1 & X_i \geq t \\ 0 & X_i < t \end{cases}$$

We denote by $Z = \frac{1}{n} \sum_{i \in [n]} Z_i$. Then, we have for the empirical measure $\mu_n$,

$$\Pr_{X \sim \mu_n}[X \geq t] = Z$$

Furthermore, we have by Definition 5:

$$\Pr[Z_i = 1] \leq 2 \exp\left(-\frac{t^2}{\sigma^2}\right)$$

Then, by Lemma 30, we have:

$$\Pr\left[Z \geq 2 \exp\left(-\frac{t^2}{\sigma^2}\right)\right] \leq \Pr\left[Z \geq 3 \exp\left(-\frac{t^2}{\sigma^2}\right)\right] \leq \exp\left(-\frac{n \exp\left(-\frac{t^2}{\sigma^2}\right)}{6}\right) \leq \delta$$

By setting $c = 6$, we get the desired result. $\square$

Now, we prove Theorem 12, restated here for ease of reading:

**Theorem 12** (Empirical Distribution is sub-Weibull). *Given an $\alpha$-sub-Weibull true measure $\mu$, the empirical measure $\mu_n$ generated by $X_1, X_2, ..., X_n$ drawn i.i.d. from $\mu$ is $(\tau, \alpha)$-sub-Weibull with probability at least $1 - \delta$ given $n \geq \frac{\exp\left(\sqrt[\alpha]{\tau}\right)}{12} \log \frac{1}{\delta}$.*

*Proof.* To capture the tail behaviour w.r.t. some $t \leq \tau$ for each sample, we define random variables $Z_1, Z_2, ..., Z_n$ as:

$$Z_i = \begin{cases} 1 & X_i \geq t \\ 0 & X_i < t \end{cases}$$

We denote by $Z = \frac{1}{n} \sum_{i \in [n]} Z_i$. Then, we have for the empirical measure $\mu_n$,

$$\Pr_{X \sim \mu_n} [X \geq t] = Z$$

Furthermore, we have:

$$\Pr[Z_i = 1] \leq c_\alpha \exp\left(-t^{1/\alpha}\right)$$

Then, by Lemma 30, we have:

$$\Pr\left[Z \geq 1.5 c_\alpha \exp\left(t^{1/\alpha}\right)\right] \leq \exp\left(-\frac{n c_\alpha \exp\left(-t^{1/\alpha}\right)}{12}\right) \leq \exp\left(-\frac{n c_\alpha \exp\left(-\sqrt[\alpha]{\tau}\right)}{12}\right) \leq \delta$$

$\square$

Next, we prove the Theorem 17. For that purpose, we need the following lemma.

**Lemma 31** (DKW Inequality (Dvoretzky et al., 1956)). *Given a sequence of i.i.d. random variables $X_1, X_2, ..., X_n$ generated from a distribution $\mu$ with true cdf $Q_\mu$. Let $Q_{\mu_n}$ be the cdf of the empirical measure $\mu_n$ generated from the samples $X_1, X_2, ..., X_n$. Then, for all $\varepsilon \geq 0$, given $n \geq \frac{\ln(2)}{2\epsilon^2}$, we have:*

$$\mathbb{P}\left(\sup_{x \in \mathbb{R}} |Q_\mu(x) - Q_{\mu_n}(x)| \geq \varepsilon\right) \leq 2\exp\left(-2n\varepsilon^2\right)$$

**Theorem 17** (Bucketed Empirical Measure is Bi-Lipschitz). *Let the bucketed empirical measure $\mu_{b,n}$ generated by $n$ i.i.d. samples coming from a distribution $\mu$ with $\ell_\mu$ bi-Lipschitz CDF. For any $\varepsilon, \delta \in (0, 1)$ and fixed constant $c > 0$, if $n \geq \frac{c}{\varepsilon^2 b^2} \log\left(\frac{2}{\delta}\right) \max\left\{\frac{1}{\ell_\mu^2}, \ell_\mu^2\right\}$, $\mu_{b,n}$ is $(1 + \varepsilon)\ell_\mu$ bi-Lipschitz with probability $1 - \delta$.*

*Proof.* For $n \geq \log\left(\frac{2}{\delta}\right) \max\left(\frac{2}{\varepsilon^2 b^2 \ell_\mu^2}, \frac{8\ell_\mu^2}{\varepsilon^2 b^2}\right)$, we have by Lemma 31:

$$\mathbb{P}\left[\sup_{x \in \mathcal{B}} |Q_\mu(x) - Q_{\mu_{b,n}}(x)| \geq \min\left(\frac{\varepsilon b \ell_\mu}{2}, \frac{\varepsilon b}{4\ell_\mu}\right)\right] \leq \delta$$

Let us denote $\varepsilon_{DKW} = \min\left(\frac{\varepsilon b \ell_\mu}{2}, \frac{\varepsilon b}{4\ell_\mu}\right)$. Then, we have with probability $1 - \delta$,

$$\sup_{x \in \mathcal{B}} |Q_\mu(x) - Q_{\mu_{b,n}}(x)| \leq \varepsilon_{DKW} . \tag{9}$$

Now, we have $\forall a, b \in \mathcal{B}$, with probability $1 - \delta$:

$$\begin{aligned} &\left|Q_{\mu_{b,n}}(a) - Q_{\mu_{b,n}}(b)\right| \\ &\leq |Q_\mu(a) - Q_\mu(b)| + 2\varepsilon_{DKW} \\ &\leq \ell_\mu |a - b| + \varepsilon \ell_\mu |a - b| \\ &\leq \ell_\mu (1 + \varepsilon) |a - b| \end{aligned}$$

Where the first inequality follows from Equation (9) and triangle inequality, and the second inequality follows from the fact that $\varepsilon_{DKW} \leq \frac{\varepsilon b \ell_\mu}{2}$ and $b \leq |a - b|, \forall a, b \in \mathcal{B}$. For the other side of the inequality,

$$
\begin{aligned}
\left| Q_{\mu_{b,n}}(a) - Q_{\mu_{b,n}}(b) \right| \\
\geq |Q_\mu(a) - Q_\mu(b)| - 2\varepsilon_{DKW} \\
\geq \frac{1}{\ell_\mu} ||a - b|| - \frac{\varepsilon |a - b|}{2\ell_\mu} \\
\geq \frac{1}{\ell_\mu (1 + \varepsilon)} |a - b|
\end{aligned}
$$

Where the first inequality follows from Equation (9) and triangle inequality, and the second inequality follows from the fact that $\varepsilon_{DKW} \leq \frac{\varepsilon b}{4\ell_\mu}$ and $b \leq |a - b|, \forall a, b \in \mathcal{B}$, and the third inequality follows from the fact that for all $\varepsilon \in (0, 1)$, $1 - \frac{\varepsilon}{2} \geq \frac{1}{1+\varepsilon}$. $\qquad \square$

## C    Estimation Guarantees of `SPA` and `SCA`

In this section, we provide the proofs for Theorem 14 and 16. We restate the theorems here for ease of reading:

**Theorem 14** (`SPA` Learning Error). *(a) If `SPA` uses $k$ buckets and outputs $\hat{\mu}_n$, then for all $i \in I$,*

$$
\left| p_{\hat{\mu}_n}(i) - p_{\mu_{b,n}}(i) \right| \leq \max \left\{ \frac{4 \text{ res}(n, k/4) \, f_i}{n}, \frac{f_i - \hat{f}_i}{n} \right\} .
$$

*(b) Further, if $\mu$ corresponding to $\mu_n$ is $\sigma_\mu$-sub-Gaussian, $|\zeta| = n \geq \frac{c}{\varepsilon} \log \left( \frac{1}{\delta} \right)$, and $k \geq \left\lceil \frac{8\sigma_\mu}{b} \sqrt{\log \left( \frac{1}{\varepsilon} \right)} \right\rceil$, then for all $i \in I$, $\mathbb{P}(\left| p_{\hat{\mu}_n}(i) - p_{\mu_{b,n}}(i) \right| \geq \varepsilon) \leq \delta$.*

*(c) Further, if $\mu$ corresponding to $\mu_n$ is $\alpha$-SubWeibull, $|\zeta| = n \geq \frac{c}{\varepsilon} \log \left( \frac{1}{\delta} \right)$, and $k \geq \left\lceil \frac{c}{b} \left( \log \left( \frac{1}{\varepsilon} \right) \right)^\alpha \right\rceil$, then for all $i \in I$, $\mathbb{P}(\left| p_{\hat{\mu}_n}(i) - p_{\mu_{b,n}}(i) \right| \geq \varepsilon) \leq \delta$.*

*Proof of Theorem 14.* **Proof of (a):** We Start with the first equation:

$$
\begin{aligned}
\hat{p}(i) - p_{\mu_{b,n}}(i) = \frac{\hat{f}_i}{\max_j \hat{F}_j} - \frac{f_i}{n} &\leq \frac{\hat{f}_i}{n - 2 \text{ res}(n, k/4)} - \frac{f_i}{n} \\
&\leq \frac{f_i}{n - 2 \text{ res}(n, k/4)} - \frac{f_i}{n} \\
&= f_i \left( \frac{1}{n - 2 \text{ res}(n, k/4)} - \frac{1}{n} \right) \\
&= \frac{f_i}{n} \left( \frac{2 \text{ res}(n, k/4)}{n - 2 \text{ res}(n, k/4)} \right) \\
&\leq \frac{f_i}{n} \frac{4 \text{ res}(n, k/4)}{n}
\end{aligned}
$$

For the second part,

$$
p_{\mu_{b,n}}(i) - \hat{p}(i) = \frac{f_i}{n} - \frac{\hat{f}_i}{\max_j \hat{F}_j} \leq \frac{f_i}{n} - \frac{\hat{f}_i}{n} \leq \frac{f_i - \hat{f}_i}{n}
$$

Finally, we have:

$$
\left| p_{\mu_{b,n}}(i) - \hat{p}(i) \right| = \max \left( p_{\mu_{b,n}}(i) - \hat{p}(i), \hat{p}(i) - p_{\mu_{b,n}}(i) \right) \leq \max \left( \frac{4 \text{ res}(n, k/4) \, f_i}{n}, \frac{f_i - \hat{f}_i}{n} \right)
$$

Now, we also have by Corollary 29 and the fact that $f_i \leq n$,

$$\left| p_{\mu_{b,n}}(i) - \hat{p}(i) \right| \leq \frac{4 \operatorname{res}(n, k/4)}{n}$$

**Proof of (b):** If $\mu$ is $\sigma_\mu$-sub-Gaussian, by Lemma 11, we know that $\mu_n$ generated by $\mu$ is $(\tau, \sigma_\mu)$ sub-Gaussian given $n \geq c \exp\left(\frac{\tau^2}{\sigma_\mu^2}\right) \log\left(\frac{1}{\delta}\right)$. Here, we can fix $\tau = 2\sigma_\mu \sqrt{\log\left(\frac{1}{\varepsilon}\right)}$. Hence, by the property of sub-Gaussian distributions 7 and the fact that each bucket of size $b$,

$$\frac{\operatorname{res}(n, k/4)}{n} = \mathbb{P}_{\mu_n}\left[X \geq \left\lceil 2\sigma_\mu \sqrt{\log\left(\frac{1}{\varepsilon}\right)} \right\rceil\right] \leq \varepsilon.$$

**Proof of (c):** For $\alpha$-sub-Weibull distributions, by Lemma 12, we know that $\mu_n$ generated by $\mu$ is $\tau, \alpha$-sub-Weibull given $n \geq \frac{\exp\left(\sqrt[\alpha]{\tau}\right)}{12} \log \frac{1}{\delta}$. Here, we can fix $\tau = c\left(\log\left(\frac{1}{\varepsilon}\right)\right)^\alpha$, and thus $n \geq \frac{1}{3\varepsilon} \log \frac{1}{\delta}$ suffices. Hence, by the property of sub-Weibull distributions (Definition 6) and the fact that each bucket of size $b$,

$$\frac{\operatorname{res}(n, k/4)}{n} = \mathbb{P}_{\mu_n}\left[X \geq \left\lceil c\left(\log\left(\frac{1}{\varepsilon}\right)\right)^\alpha \right\rceil\right] \leq \varepsilon$$

$\square$

**Theorem 16** (SCA Learning Error). *Given stream length $|\zeta| = n \geq \tau_n(\varepsilon, \delta)$ generated from a bounded-tail distribution $\mu$, SCA with bucket width $b$ sets #buckets to $k \geq \tau_k(\varepsilon, b)$ and outputs $\widehat{Q}$, such that with probability at least $1 - \delta$, $\left| Q_{\mu_{b,n}}(i) - \widehat{Q}(i) \right| \leq \varepsilon$ for all $i \in I$.*

| Tail Condition | $\tau_n(\varepsilon, \delta)$ | $\tau_k(\varepsilon, b)$ |
|---|---|---|
| $\sigma_\mu$-sub-Gaussian | $\frac{c}{\varepsilon} \log\left(\frac{1}{\delta}\right)$ | $\left\lceil \frac{8\sigma_\mu}{b} \sqrt{\log\left(\frac{4}{\varepsilon}\right)} \right\rceil$ |
| $\alpha$-sub-Weibull | $\frac{c}{\varepsilon} \log\left(\frac{1}{\delta}\right)$ | $\left\lceil \frac{c}{2b}\left(\log\left(\frac{4}{\varepsilon}\right)\right)^\alpha \right\rceil$ |

*Proof of Theorem 16.* **Step 1: Bounding the residuals in frequency estimation under tail conditions.** For sub-Gaussian distributions, by Lemma 11, we know that $\mu_n$ generated by $\mu$ is $(\tau, \sigma_\mu)$ sub-Gaussian given $n \geq c \exp\left(\frac{\tau^2}{\sigma_\mu^2}\right) \log\left(\frac{1}{\delta}\right)$. Here, we can fix $\tau = 2\sigma_\mu \sqrt{\log\left(\frac{4}{\varepsilon}\right)}$. Hence, by the property of sub-Gaussian distributions 7 and the fact that each bucket of size $b$,

$$\frac{\operatorname{res}(n, k/4)}{n} = \mathbb{P}_{\mu_n}\left[X \geq \left\lceil 2\sigma_\mu \sqrt{\log\left(\frac{4}{\varepsilon}\right)} \right\rceil\right] \leq \frac{\varepsilon}{4} \tag{10}$$

For $\alpha$-sub-Weibull distributions, by Lemma 12, we know that $\mu_n$ generated by $\mu$ is $(\tau, \alpha)$-sub-Weibull given $n \geq \frac{\exp\left(\sqrt[\alpha]{\tau}\right)}{12} \log \frac{1}{\delta}$. Here, we can fix $\tau = \left(\log\left(\frac{4}{\varepsilon}\right)\right)^\alpha$, and thus $n \geq \frac{1}{3\varepsilon} \log \frac{1}{\delta}$ suffices. Hence, by the property of sub-Weibull distributions (Definition 6) and the fact that each bucket of size $b$,

$$\frac{\operatorname{res}(n, k/4)}{n} = \mathbb{P}_{\mu_n}\left[X \geq \left\lceil c\left(\log\left(\frac{4}{\varepsilon}\right)\right)^\alpha \right\rceil\right] \leq \frac{\varepsilon}{4} \tag{11}$$

**Step 2: Bounding the error in CDF estimation.**

*Part i.* Now, we have:

$$
\begin{aligned}
Q_{\mu_{b,n}}(i) - \widehat{Q}(i) = \frac{F_i}{N} - \frac{\hat{F}_i}{\max_j \hat{F}_j} & \\
\leq \frac{F_i - \hat{F}_i}{n} && \text{since } n \geq \max_j \hat{F}_j \\
= \frac{2 \operatorname{res}(n, k/4)}{n} && \text{By Lemma 15} \\
\leq \varepsilon && \text{By Equation (10) and (11)}
\end{aligned}
$$

*Part ii.* Now, we look into the other side of the error. First, we observe that:

$$
\frac{\operatorname{res}(n, k/4)}{n - 2 \operatorname{res}(n, k/4)} \leq \frac{2 \operatorname{res}(n, k/4)}{n} \leq \varepsilon
$$

Here, the first inequality is given by the fact $2\operatorname{res}(n, k/4) \leq n$, and the second inequality follows from (10). Thus,

$$
\begin{aligned}
\widehat{Q}(i) - Q_{\mu_{b,n}}(i) = \frac{\hat{F}_i}{\max_j \hat{F}_j} - \frac{F_i}{N} & \\
\leq \frac{F_i}{\max_j \hat{F}_j} - \frac{F_i}{N}, && \text{since } F_i \geq \hat{F}_i \\
= F_i \left( \frac{1}{\max_j \hat{F}_j} - \frac{1}{n} \right) & \\
\leq F_i \left( \frac{1}{n - 2 \operatorname{res}(n, k/4)} - \frac{1}{n} \right), && \text{since } \max_j \hat{F}_j \geq \max_j F_j - 2 \operatorname{res}(n, k/4) = n - \operatorname{res}(n, k/4) \\
= F_i \left( \frac{2 \operatorname{res}(n, k/4)}{n (n - 2 \operatorname{res}(n, k/4))} \right) & \\
\leq \frac{2 \operatorname{res}(n, k/4)}{n - 2 \operatorname{res}(n, k/4)} && F_i \leq n \\
\leq \frac{4 \operatorname{res}(n, k/4)}{n} && n \geq 4 \operatorname{res}(n, k/4) \\
\leq \varepsilon && \text{By Equation (10) and (11)}
\end{aligned}
$$

$\square$

## D    Estimation Guarantee of `SWA`

In this section, we state the proofs for Theorem 18. We start with Lemma 32 which shows that a bucketed empirical measure $\mu_{b,n}$ is close to the underlying empirical measure $\mu_n$ in Wasserstein distance given the bucket size is sufficiently small.

**Lemma 32.** *Given a distribution $\mu$ and corresponding $\mu_b^{Disc}$ with bucket size $b$, we have:*

$$
\mathcal{W}_{\mathrm{p}} \left( \mu, \mu_b^{Disc} \right) \leq b
$$

*Proof.* For any $i$-th bucket corresponding to $\mu_b^{Disc}$, we have:

$$
Q_\mu(\mathcal{B}_i + \frac{b}{2}) = Q_{\mu_b^{Disc}}(\mathcal{B}_i + \frac{b}{2})
$$

$$\left| Q_\mu^{-1}(x) - Q_{\mu_b^{Disc}}^{-1}(x) \right| \le b$$

Hence, we have:

$$\mathcal{W}_{\mathrm{p}}\left(\mu, \mu_b^{Disc}\right) = \left(\int_0^1 \left| Q_\mu^{-1}(x) - Q_{\mu_b^{Disc}}^{-1}(x) \right|^p dx\right)^{\frac{1}{p}} \le \left(\int_0^1 b^p \, dx\right)^{\frac{1}{p}} = b$$

This concludes the proof. □

A general version of Lemma 32 is given in Staib et al. (2017, Theorem 4.1). However, we prove our version here for simplicity and ease of access.

Next, we prove the following lemma that characterizes the approximation of a function based on an approximation guarantee on its inverse.

**Lemma 33** (Inverse Approximation of inverse $\ell$-Lipschitz functions). *Let $f$ be a $\ell$ bi-Lipschitz function. Given an estimate $\hat{f}$ of $f$ such that $\hat{f}(x) \in [f(x) - \varepsilon, f(x) + \varepsilon]$, we can construct an estimate $\hat{f}^{-1}$ of the pseudoinverse function $f^{-1}$ such that $\forall x \in \mathrm{Range}(f)$:*

$$\hat{f}^{-1}(x) \in [f^{-1}(x) - \ell\varepsilon, f^{-1}(x) + \ell\varepsilon]$$

*Proof.* We construct $\hat{f}^{-1}$ as $\hat{f}^{-1}(y) = \min_x \left\{ x \in \mathbb{R} \cup \{-\infty\} : \hat{f}(x) = y \right\}$. Let us consider $x \in \mathrm{Range}(f)$, and $f^{-1}(x) = y_1$ and $\hat{f}^{-1}(x) = y_2$, and w.l.o.g. assume $y_2 \ge y_1$. Then, we have:

$$f(y_1) = \hat{f}(y_2)$$
$$\hat{f}(y_1) + \varepsilon \ge \hat{f}(y_2)$$
$$\varepsilon \ge \hat{f}(y_2) - \hat{f}(y_1)$$

Next, we use the fact that the function $f$ is $\ell$ bi-Lipschitz to show that $y_2 - y_1$ is bounded by $\ell\varepsilon$:

$$\begin{aligned} &y_2 - y_1 \\ \le& \hat{f}^{-1}(\hat{f}(y_2)) - \hat{f}^{-1}(\hat{f}(y_1)) \qquad && \text{By definition of } \hat{f}^{-1} \\ \le& \ell(\hat{f}(y_2) - \hat{f}(y_1)) \le \ell\varepsilon \end{aligned}$$

Hence, for any $x \in \mathrm{Range}(f)$, we have $|\hat{f}^{-1}(x) - f^{-1}(x)| \le \ell\varepsilon$, completing our proof. □

The following corollary follows directly.

**Corollary 34** (From CDF to Inverse CDF). *For an $\ell$ bi-Lipschitz distribution $\mu$ with CDF $Q_\mu$ and pseudoinverse CDF $Q_\mu^{-1}$, given an approximate CDF $\widehat{Q}_\mu$ satisfying $|\widehat{Q}_\mu(x) - Q_\mu(x)| \le \varepsilon, \forall x \in \mathbb{R}$, we construct an approximation of pseudoinverse CDF $\widehat{Q_\mu^{-1}}$, such that $\forall x \in [0, 1]$, $|\widehat{Q_\mu^{-1}}(x) - Q_\mu^{-1}(x)| \le \ell\varepsilon$.*

Now, we prove the Theorem 18. We restate the theorems here for ease of reading:

**Theorem 18** (Universal Learning of `SCA` in $\mathcal{W}_{\mathrm{p}}(,)$). *Given $|\zeta| = n \ge \tau_n\left(\ell_\mu, \varepsilon, \delta\right)$ generated from a bounded-tail distribution $\mu$ and $\varepsilon, \delta \in (0, 1)$, if we set the bucket width $b = \varepsilon/2$, then `SCA` uses $\tau_k\left(\ell_\mu, \varepsilon\right)$ buckets (i.e. space) and outputs $\hat{\mu}_n$, such that $\mathbb{P}(\mathcal{W}_{\mathrm{p}}\left(\mu_n, \hat{\mu}_n\right) \ge \varepsilon) \le \delta$. We omit the constants in $\tau_n\left(\ell_\mu, \varepsilon, \delta\right)$ and $\tau_k\left(\ell_\mu, \varepsilon\right)$ for simplicity.*

| Tail Condition | $\tau_n\left(\ell_\mu, \varepsilon, \delta\right)$ | $\tau_k\left(\ell_\mu, \varepsilon\right)$ |
|---|---|---|
| $\sigma_\mu$-sub-Gaussian | $\log\left(1/\delta\right) \max\left\{\frac{1}{\varepsilon}, \frac{1}{\ell_\mu^2}, \ell_\mu^2\right\}$ | $\frac{\sigma_\mu}{\varepsilon} \log\left(\frac{\ell_\mu}{\varepsilon}\right)$ |
| $\alpha$-sub-Weibull | $\log\left(1/\delta\right) \max\left\{\frac{1}{\varepsilon}, \frac{1}{\ell_\mu^2}, \ell_\mu^2\right\}$ | $\frac{1}{\varepsilon}\left(\log\left(\frac{\ell_\mu}{\varepsilon}\right)\right)^\alpha$ |

*Proof of Theorem 18.* By Theorem 17, 16, and Lemma 33; and the parameters we have used, we have with probability $1 - \delta$:

$$\left| Q_{\mu_{\varepsilon/2,n}}^{-1} - \widehat{Q^{-1}} \right| \leq 2\ell_\mu \quad \left| Q_{\mu_{\varepsilon/2,n}} - \widehat{Q} \right| \leq \varepsilon/2$$

Hence, by Lemma 2, we have with probability at least $1 - \delta$:

$$\mathcal{W}_{\mathrm{p}}\left(\mu_n, \hat{\mu}_n\right) \leq \mathcal{W}_{\mathrm{p}}\left(\mu_n, \mu_{\varepsilon/2,n}\right) + \mathcal{W}_{\mathrm{p}}\left(\mu_{\varepsilon/2,n}, \hat{\mu}_n\right)$$

$$\leq \varepsilon/2 + \left( \int_0^1 |Q_{\mu_{\varepsilon/2,n}}^{-1}(r) - \widehat{Q^{-1}}(r)|^p \, dr \right)^{1/p}$$

$$\leq \varepsilon/2 + \left( \int_0^1 |\varepsilon/2|^p \, dr \right)^{1/p} = \varepsilon$$

The bound on number of buckets can be obtained by plugging in the values in the line 3 of Algorithm 3 in Theorem 16. □

We now proof the Theorem 19, restated here for ease of reading:

**Theorem 19** (PAC Guarantee of `SWA`). *Given $|\zeta| = n \geq \tau_n\left(\ell_\mu, \varepsilon, \delta\right)$ generated from a bounded-tail distribution $\mu$, with bucket width $b = \varepsilon/2$, then `SCA` uses $\tau_k\left(\ell_\mu, \varepsilon\right)$ space and outputs $\hat{\mu}_n$ such that $\mathbb{P}(\mathcal{W}_1\left(\mu, \hat{\mu}_n\right) \leq \varepsilon) \geq 1 - \delta$. This further shows that for a stream of size $n = \mathcal{O}\left(\epsilon^{-2}\log(1/\delta)\right)$ from two distributions $\mu$ and $\nu$, `SWA` yields*

$$\left| \mathcal{W}_1\left(\hat{\mu}_n, \hat{\nu}_n\right) - \mathcal{W}_1\left(\mu, \nu\right) \right| \leq 4\varepsilon \tag{3}$$

*with probability $1 - 2\delta$. Here, $\epsilon \in (0, 1/4]$ and $\delta \in (0, 1/2]$. Here, $\sigma \triangleq \max\{\sigma_\mu, \sigma_\nu\}$, $\ell \triangleq \max\{\ell_\mu, \ell_\nu\}$. We omit the constants in $\tau_n\left(\ell_\mu, \varepsilon, \delta\right)$ and $\tau_k\left(\ell_\mu, \varepsilon\right)$ for simplicity.*

| Tail Condition | $\tau_n\left(\ell_\mu, \varepsilon, \delta\right)$ | $\tau_k\left(\ell_\mu, \varepsilon\right)$ |
|---|---|---|
| $\sigma$-sub-Gaussian | $\log\left(1/\delta\right) \max\left\{\frac{1}{\varepsilon^2}, \frac{1}{\ell^2}, \ell^2\right\}$ | $\frac{\sigma}{\varepsilon}\log\left(\frac{\ell}{\varepsilon}\right)$ |
| $\alpha$-sub-Weibull | $\log\left(1/\delta\right) \max\left\{\frac{1}{\varepsilon^2}, \left(\log\frac{1}{\delta}\right)^{2\alpha-1}, \frac{1}{\ell^2}, \ell^2\right\}$ | $\frac{1}{\varepsilon}\left(\log\left(\frac{\ell}{\varepsilon}\right)\right)^\alpha$ |

*Proof.* We combine the results of Corollary 44, 45, and Theorem 18 to establish this result. By Corollary 44 for sub-Gaussian distributions, we have with probability $1 - 2\delta$:

$$|\mathcal{W}_1\left(\mu, \nu\right) - \mathcal{W}_1\left(\mu_n, \nu_n\right)| \leq 2\varepsilon \qquad \text{By Triangle Inequality} \tag{12}$$

Similarly, by Corollary 45 for $\alpha$-sub-Weibull distributions, we have with probability $1 - 2\delta$:

$$|\mathcal{W}_1\left(\mu, \nu\right) - \mathcal{W}_1\left(\mu_n, \nu_n\right)| \leq 2\varepsilon \qquad \text{By Triangle Inequality} \tag{13}$$

By Theorem 19, we have with probability $1 - 2\delta$:

$$|\mathcal{W}_1\left(\hat{\mu}_n, \hat{\nu}_n\right) - \mathcal{W}_1\left(\mu_n, \nu_n\right)| \leq 2\varepsilon \qquad \text{By Triangle Inequality} \tag{14}$$

A union bound argument and triangle inequality over Equations (12) or (13) and (14) completes the proof. □

## E  Estimation Guarantee of `STVA`

In this section, we prove the theorems concerning `STVA`, i.e. Theorem 21, Lemma 22, and Theorems 23 and 24.

### E.1 Proof of Theorem 21

In this section, we detail the proof of Theorem 21.

**Theorem 21** (Bucket Width for TV Distance Estimation)**.** *Given a bounded-tail distribution $\mu$ with $\ell_\mu$-Lipschitz PDF and a corresponding bucketed measure $\mu_b$, if we fix the bucket width $b = \tau_b(\varepsilon, \ell)$, we have* TV $(\mu, \mu_b) \le \epsilon$.

| Tail Condition | $\sigma_\mu$-sub-Gaussian | $\alpha$-sub-Weibull |
|---|---|---|
| $\tau_b(\varepsilon, \ell_\mu)$ | $\frac{\varepsilon}{\sigma_\mu \ell_\mu \sqrt{\log(2/\varepsilon)}}$ | $\frac{\varepsilon}{\ell_\mu (\log(2c_\alpha/\varepsilon))^\alpha}$ |

*Proof.* **Initial Bounds:** The distribution $\mu$ has a $\ell_\mu$-Lipschitz PDF. Consider any bucket $\sqcup$ of size $b$, then $\min_{y \in \sqcup} \mu(y) - \max_{y \in \sqcup} \mu(y) \le b\ell_\mu$. As $\min_{y \in \sqcup} \mu(y) \le \mu_b(x) \le \max_{y \in \sqcup} \mu(y), \forall x \in \sqcup$, we have for any bucket $\sqcup$:

$$|\mu(x) - \mu_b(x)| \le b\ell_\mu \qquad\qquad \forall x \in \sqcup$$

$$\int_\sqcup |\mu(x) - \mu_b(x)|\, \mathrm{d}x \le b^2 \ell_\mu \tag{15}$$

The main idea of our proof is that we bound the error due to bucketing in buckets with high frequency and use the fact that the remaining buckets has sufficiently small frequency to achieve our bound. To denote the buckets with high frequency, recall that $Top_k$ denotes the set of $k$ buckets with highest frequency, $b$ denotes the length of each bucket.

$$
\begin{aligned}
&\text{TV}(\mu, \mu_b) \\
=&\frac{1}{2}\int_{-\infty}^{\infty} |\mu(x) - \mu_b(x)|\,\mathrm{d}x \\
=&\frac{1}{2}\sum_{\sqcup \in \mathcal{B}} \int_\sqcup |\mu(x) - \mu_b(x)|\,\mathrm{d}x \\
=&\frac{1}{2}\sum_{\sqcup \in Top_k} \int_\sqcup |\mu(x) - \mu_b(x)|\,\mathrm{d}x + \frac{1}{2}\sum_{\sqcup \notin Top_k} \int_\sqcup |\mu(x) - \mu_b(x)|\,\mathrm{d}x \\
\le&\frac{1}{2}\sum_{\sqcup \in Top_k} b^2 \ell_\mu + \frac{1}{2}\sum_{\sqcup \notin Top_k} \left[\int_\sqcup \mu_b(x)\mathrm{d}x + \int_\sqcup \mu(x)\mathrm{d}x\right] \qquad \text{By Equation (15)} \\
\le&kb^2 \ell_\mu/2 + \sum_{\sqcup \notin Top_k} \int_\sqcup \mu(x)\mathrm{d}x
\end{aligned}
\tag{16}
$$

$$\tag{17}$$

**Sub-Weibull Distributions:** For $\alpha$-sub-Weibull distributions, we fix $k = \frac{\ell_\mu (\log(2c_\alpha/\varepsilon))^{2\alpha}}{\varepsilon}$ and bucket size $b = \frac{\varepsilon}{\ell_\mu (\log(2c_\alpha/\varepsilon))^\alpha}$ and bound the terms in Equation (17). For the first term, we have:

$$kb^2 \frac{\ell_\mu}{2} = \frac{\ell_\mu (\log(2c_\alpha/\varepsilon))^{2\alpha}}{\varepsilon} \left(\frac{\varepsilon}{\ell_\mu (\log(2c_\alpha/\varepsilon))^\alpha}\right)^2 \frac{\ell_\mu}{2} \le \varepsilon/2 \tag{18}$$

For the second term:

$$\sum_{\sqcup \notin Top_{\frac{\ell_\mu(\log(2/\varepsilon))^{2\alpha}}{\varepsilon}}} \int_\sqcup \mu(x)\mathrm{d}x = \mathop{\mathbb{P}}_{X \sim \mu}\left[X \in \cup_{\sqcup \notin Top_{\frac{\ell_\mu(\log(2/\varepsilon))^{2\alpha}}{\varepsilon}}} \sqcup\right]$$

$$\le \mathop{\mathbb{P}}_{X \sim \mu}\left[X \ge \left(\log \frac{2c_\alpha}{\varepsilon}\right)^\alpha\right] \le \varepsilon/2 \tag{19}$$

Combining Equations (18) and (19) with the bound from Equation (17), we obtain the result.

**Sub-Gaussian Distributions:** For sub-Gaussian distributions, we fix $k = \frac{\sigma_\mu^2 \ell_\mu \log(2/\varepsilon)}{\varepsilon}$ and bucket size $b = \frac{\varepsilon}{\sigma_\mu \ell_\mu \sqrt{\log(2/\varepsilon)}}$ and bound the terms in Equation (17). For the first term, we have

$$kb^2 \frac{\ell_\mu}{2} = \frac{\sigma_\mu^2 \ell_\mu \log(2/\varepsilon)}{\varepsilon} \left( \frac{\varepsilon}{\sigma_\mu \ell_\mu \sqrt{\log(2/\varepsilon)}} \right)^2 \frac{\ell_\mu}{2} \le \varepsilon/2 \tag{20}$$

For the second term:

$$\sum_{\sqcup \notin Top_{\frac{\sigma_\mu^2 \ell_\mu \log^2(2/\varepsilon)}{\varepsilon}}} \int_\sqcup \mu(x)\mathrm{d}x = \mathop{\mathbb{P}}_{X \sim \mu} \left[ X \in \cup_{\sqcup \notin Top_{\frac{\sigma_\mu^2 \ell_\mu \log(2/\varepsilon)}{\varepsilon}}} \sqcup \right]$$

$$\le \mathop{\mathbb{P}}_{X \sim \mu} \left[ X \ge \sigma_\mu \sqrt{\log(2/\varepsilon)} \right]$$

$$\le \varepsilon/2 \tag{21}$$

Combining Equations (20) and (21) with the bound from Equation (17), we obtain the result. $\qquad\square$

## E.2 Proof of Lemma 22

Consider the two possible ways of looking at a histogram. Given a histogram defined over $\mathbb{R}$, one can think of it as defining a (not necessarily probability) measure over $\mathbb{R}$ defined so that the measure at each point is equal to the frequency of the bucket the point lies in. Alternatively, one can define a measure over the buckets with the measure at each bucket is equal to the frequency of the bucket. In this section, we formalize these notions and establish the relation between them in term of TV and Wasserstein distances.

**Definition 35** (Bucketed Continuous Distribution). Let $\mathcal{D}$ be a distribution supported on a (finite or infinite) closed interval $I \subseteq \mathbb{R}$ with (discrete or continuous) measure $\mu$. Given $x_0$ as a reference point in $I$ and $b$ as the bucket width, we further represent $I$ as $I(x_0, J, b) \triangleq \cup_{j \in J}[x_0 + jb, x_0 + (j+1)b]$. Here, index set $J \subseteq \mathbb{Z}$. We define the corresponding bucketed continuous distribution $\mathcal{D}_b^{Cont}$ with measure $\mu_b^{Cont}$ defined over $I \subseteq \mathbb{R}$ as:

$$\mu_b^{Cont}(x) = \frac{1}{b} \int_{x_0+jb}^{x_0+(j+1)b} \mu(x)\mathrm{d}x$$
$$= \frac{1}{b} \mu([x_0 + jb, x_0 + (j+1)b]),$$

where $x \in [x_0 + jb, x_0 + (j+1)b]$.

**Definition 36** (Bucketed Discrete Distribution). Let $\mathcal{D}$ be a distribution supported on a (finite or infinite) closed interval $I \subseteq \mathbb{R}$ with (discrete or continuous) measure $\mu$. Given $x_0$ as a reference point in $I$ and $b$ as the bucket width, we further represent $I$ as $I(x_0, J, b) \triangleq \cup_{j \in J}[x_0 + jb, x_0 + (j+1)b]$. We define the corresponding bucketed discrete distribution $\mathcal{D}_b^{Disc}$ with measure $\mu_b^{Disc}$ defined over $\cup_{j \in J}\{x_0 + jb + b/2\} \subseteq \mathbb{R}$ as $\mu_b^{Disc}(j) = \int_{x_0+jb}^{x_0+(j+1)b} \mu(x)\mathrm{d}x = \mu([x_0 + jb, x_0 + (j+1)b])$.

The support set of $\mu_b^{Disc}$ inherits the metric structure of $\mathbb{R}$. For ease of notation, we express the distance between the $i$-th and $j$-th support points of $\mu_b^{Disc}$ as $d(\sqcup_i, \sqcup_j) \triangleq |i - j|b$.

We introduce the following lemma establishing the equivalence of bucketed discrete and continuous measure with respect to the TV distance.

**Lemma 37** (TV of continuous and discrete bucketed distributions). *Given two measures $\mu$, and $\nu$, we have:*

$$\mathrm{TV}\left(\mu_b^{Cont}, \nu_b^{Cont}\right) = \mathrm{TV}\left(\mu_b^{Disc}, \nu_b^{Disc}\right)$$

*Proof.*

$$
\begin{aligned}
\text{TV}\left(\mu_b^{Cont}, \nu_b^{Cont}\right) =& \frac{1}{2} \int \left|\mu_b^{Cont}(x) - \nu_b^{Cont}(x)\right| \mathrm{d}x \\
=& \frac{1}{2} \sum_{j \in J} \frac{1}{b} \int_{x_0+jb}^{x_0+(j+1)b} \left|\int_{x_0+jb}^{x_0+(j+1)b} \mu(x)\mathrm{d}x - \int_{x_0+jb}^{x_0+(j+1)b} \nu(x)\mathrm{d}x\right| \mathrm{d}x \\
=& \frac{1}{2} \sum_{j \in J} \left|\int_{x_0+jb}^{x_0+(j+1)b} \mu(x)\mathrm{d}x - \int_{x_0+jb}^{x_0+(j+1)b} \nu(x)\mathrm{d}x\right| \\
=& \text{TV}\left(\mu_b^{Disc}, \nu_b^{Disc}\right)
\end{aligned}
$$

$\square$

We now establish a TV distance concentration result on bucketed empirical distribution corresponding to a sub-Gaussian distribution.

**Lemma 22** (Concentration in TV over Infinite Buckets)**.** *Let $\mu_n$ be an empirical measure generated from a discrete bucketed measure $\mu_b$ corresponding to a bounded-tail distribution $\mu$, and, $\varepsilon, \delta \in (0,1)$. Then, for $n \geq \tau_n(\varepsilon, b, \delta)$, $\mathbb{P}\left[\text{TV}(\mu, \mu_n) \geq \varepsilon\right] \leq \delta$. Here, $\Gamma(\cdot)$ denotes the Gamma function (Davis, 1959).*

| Tail Condition | $\sigma_\mu$-sub-Gaussian | $\alpha$-sub-Weibull |
|---|---|---|
| $\tau_n(\varepsilon, b, \delta)$ | $c\varepsilon^{-2} \max\left\{\frac{4\sigma_\mu\sqrt{\pi}}{b}, \log(1/\delta)\right\}$ | $c\varepsilon^{-2} \max\left\{\frac{2c_\alpha}{b}\Gamma(1+\alpha), \log(1/\delta)\right\}$ |

*Proof.* Let us consider the standard bijection from $\mathbb{Z}$ to $\mathbb{N}$ as:

$$
f(x) = \begin{cases} 2\text{x} & \text{if } x > 0 \\ \text{-2x+1} & \text{if } x \leq 0 \end{cases}
$$

$\sigma_\mu$**-Sub-Gaussian Distributions:** For sub-Gaussian distributions, If $i$ is even, we have:

$$
p(i) = \mu_b^{Disc}(\sqcup_{i/2}) \leq \mu_b^{Disc}\left(\cup_{j \geq i/2}\sqcup_j\right) \leq \mu(\{x | x \geq bi/2\}) \leq 2\exp\left(-\frac{i^2 b^2}{4\sigma_\mu^2}\right).
$$

If $i$ is odd, we have:

$$
p(i) = \mu_b^{Disc}(\sqcup_{-(i-1)/2}) \leq \mu_b^{Disc}\left(\cup_{j \leq -(i-1)/2}\sqcup_j\right) \leq \mu(\{x | x \leq -b(i-1)/2\}) \leq 2\exp\left(-\frac{(i-1)^2 b^2}{4\sigma_\mu^2}\right).
$$

Combining these terms, we have:

$$
\sum_{i \in \mathbb{N}} \sqrt{p_\mu(i)} \leq \sum_{i \in \mathbb{N}} 4\exp\left(-i^2 b^2/4\sigma_\mu^2\right) \leq 4\int_0^\infty \exp(-x^2 b^2/\sigma_\mu^2)\mathrm{d}x = 2\sqrt{4\pi\sigma_\mu^2/b^2} = \frac{4\sigma_\mu\sqrt{\pi}}{b}.
$$

We now combine this with Lemma 42 to complete the proof.

$\alpha$**-SubWeibull Distributions:** For $\alpha$-SubWeibull distributions, if $i$ is even, we have:

$$
p(i) = \mu_b^{Disc}(\sqcup_{i/2}) \leq \mu_b^{Disc}\left(\cup_{j \geq i/2}\sqcup_j\right) \leq \mu(\{x | x \geq bi/2\}) \leq c_\alpha \exp\left(-(ib)^{1/\alpha}\right).
$$

If $i$ is odd, we have:

$$p(i) = \mu_b^{Disc}(\sqcup_{-(i-1)/2}) \leq \mu_b^{Disc}\left(\cup_{j \leq -(i-1)/2}\sqcup_j\right) \leq \mu(\{x | x \leq -b(i-1)/2\}) \leq c_\alpha \exp\left(-((i-1)b)^{1/\alpha}\right).$$

Combining these terms, we have:

$$\sum_{i \in \mathbb{N}} \sqrt{p_\mu(i)} \leq \sum_{i \in \mathbb{N}} 2c_\alpha \exp\left(-(ib)^{1/\alpha}\right) \leq 2c_\alpha \int_0^\infty \exp\left(-(xb)^{1/\alpha}\right) \mathrm{d}x = 2c_\alpha \frac{\Gamma(1+\alpha)}{b}.$$

We now combine this with Lemma 42 to complete the proof. $\qquad\square$

### E.3 Bounding Errors of SPA and STVA (Theorems 23 and 24)

**Theorem 23** (Learning Error of SPA)**.** *If SPA accesses a stream of length $|\zeta| = n \geq \tau_n(\varepsilon, \delta)$ from a bounded-tail distribution, and uses $\tau_k(\varepsilon, \delta)$ buckets to output a sublinear summary $p_{\hat{\mu}_n}$, then with probability $1 - \delta$, $\mathrm{TV}(\hat{\mu}_n, \mu_{b,n}) \leq \left\|p_{\hat{\mu}_n} - p_{\mu_{b,n}}\right\|_1 \leq \varepsilon$.*

| Tail Condition | $\tau_n(\varepsilon, \delta)$ | $\tau_k(\varepsilon, b)$ |
|---|---|---|
| $\sigma_\mu$-sub-Gaussian | $\frac{c}{\varepsilon}\log\left(\frac{1}{\delta}\right)$ | $\left\lceil \frac{8\sigma_\mu}{b}\sqrt{\log\left(\frac{6}{\varepsilon}\right)} \right\rceil$ |
| $\alpha$-sub-Weibull | $\frac{c}{\varepsilon}\log\left(\frac{1}{\delta}\right)$ | $\left\lceil \frac{c}{b}\left(\log\left(\frac{6}{\varepsilon}\right)\right)^\alpha \right\rceil$ |

*Proof.* **Step 1: Bounding the residuals in frequency estimation under tail conditions.** For sub-Gaussian distributions, by Theorem 11, we know that $\mu_{b,n}$ generated by $\mu_b^{Disc}$ is $(\tau, \sigma_\mu)$ sub-Gaussian given $n \geq c\exp\left(\frac{\tau^2}{\sigma_\mu^2}\right)\log\left(\frac{1}{\delta}\right)$. Here, we can fix $\tau = 2\sigma_\mu\sqrt{\log\left(\frac{6}{\delta}\right)}$. Hence, by the property of sub-Gaussian distributions 7 and the fact that length of each bucket is $b$,

$$\frac{\mathrm{res}(n, k/4)}{n} \leq \mathop{\mathbb{P}}_{\mu_{b,n}}\left[X \geq \left\lceil 2\sigma_\mu\sqrt{\log\left(\frac{6}{\varepsilon}\right)} \right\rceil\right] \leq \frac{\varepsilon}{6} \tag{22}$$

For $\alpha$-sub-Weibull distributions, by Lemma 12, we know that $\mu_n$ generated by $\mu$ is $(\tau, \alpha)$-sub-Weibull given $n \geq \frac{\exp\left(\sqrt[\alpha]{\tau}\right)}{12}\log\frac{1}{\delta}$. Here, we can fix $\tau = 2\left(\log\left(\frac{6}{\varepsilon}\right)\right)^\alpha$, and thus $n \geq \frac{1}{2\varepsilon}\log\frac{1}{\delta}$ suffices. Hence, by the property of sub-Weibull distributions (Definition 6) and the fact that each bucket of size $b$,

$$\frac{\mathrm{res}(n, k/4)}{n} = \mathop{\mathbb{P}}_{\mu_n}\left[X \geq \left\lceil c\left(\log\left(\frac{6}{\varepsilon}\right)\right)^\alpha \right\rceil\right] \leq \frac{\varepsilon}{4} \tag{23}$$

**Step 2: Bounding the error in PDF estimation.** Now, we proceed to bound the TV Distance. We denote by $Top_k$ the set of $k$ elements with highest true frequency $f_i$. Then, we have:

$$
\begin{aligned}
\left\| \hat{p} - p_{\mu_{b,n}} \right\|_1 &= \sum_i \left| p_{\mu_{b,n}}(i) - \hat{p}(i) \right| \\
&\leq \sum_i \max \left( \frac{4 \operatorname{res}(n, k/4)}{n} \cdot \frac{f_i}{n}, \frac{f_i - \hat{f}_i}{n} \right) && \text{Theorem 14} \\
&\leq \sum_i \frac{4 \operatorname{res}(n, k/4)}{n} \cdot \frac{f_i}{n} + \sum_i \frac{f_i - \hat{f}_i}{n} \\
&\leq \frac{4 \operatorname{res}(n, k/4)}{n} \sum_i \frac{f_i}{n} + \sum_{i \in Top_{k/4}} \frac{f_i - \hat{f}_i}{n} + \sum_{i \notin Top_{k/4}} \frac{f_i}{n} \\
&\leq \frac{4 \operatorname{res}(n, k/4)}{n} + \frac{\sum_{i \in LearningTop_{k/4}} f_i - \hat{f}_i + \sum_{i \notin Top_{k/4}} f_i}{n} \\
&\leq \frac{4 \operatorname{res}(n, k/4)}{n} + \frac{2 \operatorname{res}(n, k/4)}{n} && \text{Lemma 4} \\
&= \frac{6 \operatorname{res}(n, k/4)}{n} \\
&\leq \varepsilon && \text{Equation (22) and (23)}
\end{aligned}
$$

$\square$

We now proof the Theorem 24, restated here for ease of reading:

**Theorem 24** (PAC Guarantee of `STVA`). *Given $\epsilon \in (0, 1/6]$, $\delta \in (0, 1/4]$, a stream of size $n \geq \tau_n(\varepsilon, \ell, \delta)$ from two bounded-tail and $\ell$-Lipschitz distributions $\mu$ and $\nu$, and bucket width $b = \tau_b(\varepsilon, \ell, \delta)$, `STVA` uses $\tau_k(\varepsilon, \ell, \delta)$ space and*

$$
\mathbb{P}\left( \left| \operatorname{TV}(\hat{\mu}_n, \hat{\nu}_n) - \operatorname{TV}(\mu, \nu) \right| \geq 6\varepsilon \right) \leq 4\delta. \tag{4}
$$

*Here, $\sigma \triangleq \max\{\sigma_\mu, \sigma_\nu\}$, $\ell \triangleq \max\{\ell_\mu, \ell_\nu\}$. We omit the constants in $\tau_n(\ell_\mu, \varepsilon, \delta)$ and $\tau_k(\ell_\mu, \varepsilon)$ for simplicity.*

| Tail Condition | $\tau_n(\varepsilon, \ell, \delta)$ | $\tau_k(\varepsilon, \ell)$ | $\tau_b(\varepsilon, \ell)$ |
|---|---|---|---|
| $\sigma$-sub-Gaussian | $\varepsilon^{-2} \max\left\{ \frac{\sigma^2 \ell \log(1/\varepsilon)}{\varepsilon}, \log\left(\frac{1}{\delta}\right) \right\}$ | $\frac{\sigma^2 \ell}{\varepsilon} \log\left(\frac{1}{\varepsilon}\right)$ | $\frac{\varepsilon}{\sigma \ell \sqrt{\log(2/\varepsilon)}}$ |
| $\alpha$-sub-Weibull | $\varepsilon^{-2} \max\left\{ \frac{\ell(\log(1/\varepsilon))^\alpha}{\varepsilon} \Gamma(1+\alpha), \log\left(\frac{1}{\delta}\right) \right\}$ | $\frac{\ell}{\varepsilon} \left(\log\left(\frac{1}{\varepsilon}\right)\right)^{2\alpha}$ | $\frac{\varepsilon}{\ell(\log(2c_\alpha/\varepsilon))^\alpha}$ |

*Proof.* Composing Lemma 22, Lemma 37, Theorem 21, and Theorem 23 yields the proof.

By Theorem 21, and Lemma 37, we have:

$$
\begin{aligned}
\left| \operatorname{TV}(\mu, \nu) - \operatorname{TV}\left(\mu_b^{Cont}, \nu_b^{Cont}\right) \right| &\leq 2\varepsilon && \text{By Theorem 21 and Triangle Inequality} \\
\left| \operatorname{TV}(\mu, \nu) - \operatorname{TV}\left(\mu_b^{Disc}, \nu_b^{Disc}\right) \right| &\leq 2\varepsilon && \text{By Lemma 37} \tag{24}
\end{aligned}
$$

From Lemma 22, and fixing the $b$, we have with probability $1 - 2\delta$:

$$
\left| \operatorname{TV}(\mu_{b,n}, \nu_{b,n}) - \operatorname{TV}\left(\mu_b^{Disc}, \nu_b^{Disc}\right) \right| \leq 2\varepsilon \qquad \text{By Triangle Inequality} \tag{25}
$$

From Theorem 23, we have with probability $1 - 2\delta$:

$$
\left| \operatorname{TV}(\hat{\mu}_n, \hat{\nu}_n) - \operatorname{TV}(\mu_{b,n}, \nu_{b,n}) \right| \leq 2\varepsilon \qquad \text{By Triangle Inequality} \tag{26}
$$

A union bound argument and triangle inequality over Equation (24), (25) and (26) completes the proof. $\square$

## F   Useful Technical Results

**Lemma 38** (Inverse of Bi-Lipschitz function is Bi-Lipschitz ). *Given an invertible function $f : \text{Dom } f \to$ Range $f$ that is $\ell$ bi-Lipschitz , the corresponding inverse function $f^{-1} : \text{Range } f \to \text{Dom } f$ is also bi-Lipschitz with parameter $\ell$.*

The following corollary is directly implied from Lemma 38.

**Corollary 39** (Bi-Lipschitz CDF implies Bi-Lipschitz Inverse CDF). *A distribution satisfying Assumption 9 has bi-Lipschitz inverse CDF.*

**Definition 40** ($\alpha$-SubWeibull Distribution and Random Variable (Vladimirova et al., 2020)). A distribution $\mu$ is said to be $\alpha$-SubWeibull if there exists some constant $c_\alpha$ for any $t \geq 0$, we have:

$$\Pr_{X \sim \mu} [X \geq t] \leq c_\alpha \exp \left( -t^{1/\alpha} \right) \tag{27}$$

Correspondingly, the random variable $X$ drawn from $\mu$ is said to be a SubWeibull random variable.

**Lemma 41** (MGF Charecterization of SubWeibull(Theorem 2.1 in Vladimirova et al. (2020))). *Given $X$ be a $\alpha$-SubWeibull random variable, then the MGF of $|X|^{1/\alpha}$ satisfies:*

$$\exists c > 0 \text{ such that } \mathbb{E} \left[ \exp \left( (\gamma |X|)^{1/\alpha} \right) \right] \leq \exp \left( (\gamma c)^{1/\alpha} \right)$$

*for all $\gamma$ such that $0 < \gamma \leq 1/c$.*

**Lemma 42** (True Measure Concentration (Cohen et al., 2020)). *For a measure $\mu$ defined over a simplex $\Delta_\mathbb{N}$ and corresponding empirical distribution $\mu_n$ generated by $n$ samples, for any $\varepsilon, \delta \in (0, 1)$, there exists a constant $c$ such that $n \geq c\varepsilon^{-2} \max \left\{ \sum_{i \in \mathbb{N}} \sqrt{p(i)}, \log (1/\delta) \right\}$, we have with probability $1 - \delta$,*

$$\Pr [\text{TV} (\mu, \mu_n) \geq \varepsilon] \leq \delta$$

**Lemma 43** (Concentration of Empirical Measure In Wasserstein Distance (Theorem 2 in Fournier & Guillin (2015))). *Let $\mu$ be a distribution on $\mathbb{R}$. Then for all $p \in \mathbb{N}$,*

- *If $\exists \alpha < \frac{1}{p}, \gamma > 0$ such that $\mathbb{E} \left[ \exp \left( (\gamma |X|)^{1/\alpha} \right) \right] < \infty$, then,*

$$\mathbb{P} [\mathcal{W}_1 (\mu, \mu_n) \geq \varepsilon] \leq \exp \left( -cn\varepsilon^2 \right) + \exp \left( -cn\varepsilon^{1/p\alpha} \right)$$

- *If $\exists \alpha \in \left( \frac{1}{p}, \infty \right), \gamma > 0$ such that $\mathbb{E} \left[ \exp \left( (\gamma |X|)^{1/\alpha} \right) \right] < \infty$, then,*

$$\mathbb{P} [\mathcal{W}_1 (\mu, \mu_n) \geq \varepsilon] \leq \exp \left( -cn\varepsilon^2 \right) + \exp \left( -c (n\varepsilon)^{1/2p\alpha} \right)$$

The Lemma 43 implies the following two corollaries:

**Corollary 44** (Concentration in Wasserstein distance of Empirical Measures over $\mathbb{R}$ (Bhat & L.A., 2019)). *Given an empirical measure $\mu_n$ generated by $n$ i.i.d. samples generated from a subGaussian measure $\mu$ over $\mathbb{R}$, we have:*

$$\mathbb{P} [\mathcal{W}_1 (\mu, \mu_n) \geq \varepsilon] \leq \exp \left( -cn\varepsilon^2 \right)$$

**Corollary 45.** *Let $\mu$ be a distribution on $\mathbb{R}$ such that $\exists \alpha \in (1, \infty), \gamma > 0$ such that $\mathbb{E} \left[ \exp \left( (\gamma |X|)^{1/\alpha} \right) \right] < \infty$, then,*

$$\mathbb{P} [\mathcal{W}_1 (\mu, \mu_n) \geq \varepsilon] \leq \exp \left( -cn\varepsilon^2 \right) + \exp \left( -c (n\varepsilon)^{1/2\alpha} \right)$$

**Lemma 46** (Strong Demographic Parity and $\mathcal{W}_1(,)$ distance (Jiang et al., 2020)). *Let $f : \mathbb{R}^d \times [k] \to [0,1]$ be a function where $[k]$ denotes the sensitive attribute. Let $\mu_s$ denote the output distribution of $f$ corresponding to the sensitive attribute $s \in [k]$. Then, we have for all $s, s' \in [k]$:*

$$\underset{t \sim \mathrm{U}[0,1]}{\mathbb{E}} \left[ \mathbb{P}\left[ f(X,s) \geq t \right] - \mathbb{P}\left[ f(X,s') \geq t \right] \right] = \mathcal{W}_1\left( \mu_s, \mu_{s'} \right)$$

**Lemma 47** (Privacy and Hockey Stick Divergence(Balle et al., 2018)). *For a given $\alpha \in \mathbb{R}$, a mechanism $\mathcal{M}$ is $(\alpha, \beta)$-differentially private if for all $\mathbf{X}, \mathbf{X}'$ with $\mathrm{Ham}\left( \mathbf{X}, \mathbf{X}' \right) = 1$:*

$$\mathrm{H}_{\mathrm{e}^\alpha}\left( \mathcal{M}\left( \mathbf{X} \right) || \mathcal{M}\left( \mathbf{X}' \right) \right) \leq \beta$$

*where $\mathrm{H}_{\mathrm{e}^\alpha}\left( \mu || \nu \right)$ is called the Hockeystick divergence between $\mu$ and $\nu$.*

**Lemma 48** (From HSD to TV Approximation (Koskela & Mohammadi, 2024)). *Given two measured $\mu, \nu$ and their TV distance approximations $\hat{\mu}_n, \hat{\nu}_n$ satisfying $\mathrm{TV}\left( \mu, \hat{\mu}_n \right) \leq \varepsilon$, and $\mathrm{TV}\left( \hat{\mu}_n, \hat{\nu}_n \right) \leq \varepsilon$; we have for all $\alpha \in \mathbb{R}$:*

$$\mathrm{H}_{\mathrm{e}^\alpha}\left( \mu || \nu \right) \leq \mathrm{H}_{\mathrm{e}^\alpha}\left( \hat{\mu}_n || \hat{\nu}_n \right) + \left( 1 + e^\alpha \right) \varepsilon$$

# G    Experimental Details - Fairness and Privacy Auditing

## G.1    Fairness Auditing

Group fairness measures in ML model's prediction measure the disparity in predictions of ML models across different subpopulations. The subpopulations correspond to a sensitive attribute (e.g. gender, economic status, ethnicity etc.) on which they should not be discriminated. Fairness auditors try to estimate violations of such group fairness measures by querying a model and then collecting the responses (Ghosh et al., 2021; Yan & Zhang, 2022; Chugg et al., 2024; Ajarra et al., 2025).

A popular group fairness measure is demographic parity (Feldman et al., 2015).

**Definition 49** (Demographic Parity)**.** Given $O \subseteq \mathbb{R}$ and sensitive attributes $[k]$, a regression function $f : \mathbb{R}^d \times [k] \to O$ satisfies demographic parity if for all $s, s' \in [k]$, we have $\sup_{t \in O} \mathbb{P}(f(X, s) \leq t) - \mathbb{P}(f(X, s') \leq t) = 0$. For classification, $S = [0, 1]$ and the definition is referred as strong demographic parity (Jiang et al., 2020).

Jiang et al. (2020) show that bounding strong demographic parity is equivalent parity is equivalent to bounding the maximum of 1-Wasserstein distance between the output distributions of any two subpopulations. This motivates us to use `SWA` to estimate demographic parity for ML models.

**Experimental Setup:** We test accuracy and sublinearity of `SWA` for fairness auditing on the well-known ACS Income dataset (Ding et al., 2021). We test both on linear regression ($d = 10$) for income as output and classification with logistic regression ($d = 10$) for income above and below 40000 USD as outputs. We compute Wasserstein Distance between the distribution of outputs of linear and logistic regression models for male and female data points of a model trained on ACS_Income data. We use $3 : 1$ train-test split for both cases. We use scikit-learn (Pedregosa et al., 2011) to train both the models, and use the 'liblinear' solver for logistic regression. For reference, we compute the distance between the bucketed versions of the output distribution exactly. We choose the bucket size to be 10 and 0.01, for the regression and classification tasks, respectively. For the regression task, we increase the number of buckets as $\{500, 1000, \dots, 20000\}$; and for the classification task, we increase the number of buckets as $\{50, 100, \dots, 1250\}$. Finally we report the multiplicative approximation error of our estimates w.r.t. the true distance in both the cases with increasing number of buckets. We run each of the experiments 50 times.

**Results: Sublinearity of `SWA` in Fairness Auditing:** We use 416625 total samples consisting of almost half male and female samples each. The sample streams arrive via $S = 10$ sources. For the linear regression model, the approximation error drops below 0.1 when we use 12500 buckets. For the logistic regression model, the approximation error drops below 0.1 when we use 750 buckets. The difference in #buckets is due to the difference in variance and the width of buckets used in each case.

## G.2    Privacy Auditing

Differential privacy (Dwork, 2006) is now considered as the gold standard for data privacy protection. It aims to keep an input datum indistinguishable while looking into outputs of an algorithm.

**Definition 50** (($\alpha, \beta$)**- Differential Privacy** (Dwork, 2006))**.** An algorithm $f : \mathcal{X} \to \mathcal{Y}$ is $(\alpha, \beta)$-differentially private if for any $\mathbf{X}, \mathbf{X}'$ with $\mathrm{Ham}(\mathbf{X}, \mathbf{X}') = 1$ and $\forall S \subseteq \mathcal{Y}$, we have $\mathbb{P}[f(\mathbf{x}) \in S] \leq e^{\alpha} \mathbb{P}[f(\mathbf{x}') \in S] + \beta$.

An equivalent representation of differential privacy is $\mathrm{H}_{e^{\alpha}}(\mathcal{M}(\mathbf{X})||\mathcal{M}(\mathbf{X}')) \leq \beta$ (Balle et al., 2018), where Hockey Stick Divergence (HSD) is defined as $\mathrm{H}_{\tau}(\mu||\nu) \triangleq \int_{\mathcal{X}}[\mu(x) - \tau\nu(x)]_+ \mathrm{d}x$. Recently, Koskela & Mohammadi (2024) show that estimating the HSD of two distributions is equivalent to estimating the HSD and TV distance between their empirical counterparts. They construct histograms over outputs of a black-box auditor and use this result to estimate TV distances for privacy auditing. As auditing privacy is data intensive, it motivates us to use `STVA` in this setting.

We adopt the experimental setting of Annamalai & Cristofaro (2024) and compute the TV distance between the output distributions of logistic regressors trained on neighbouring datasets, say IN and OUT, sampled from MNIST (LeCun, 1998).

**Experimental Setup:** We study the performance of `STVA` in the case of privacy auditing. We generate losses for datasets with and without canary using the work of Annamalai & Cristofaro (2024). We run logistic regression on `MNIST` dataset with 50 epochs and $\alpha = 10$. We denote the losses with and without canary to be `lossesin` and `lossesin`, respectively. We obtain 1000 samples of losses for both the cases and take these losses as the distribution of interest. We choose the bucket size to be 0.01 for the logistic regression task. The mean of `lossesin` and `lossesout` are $-2.3194$ and $-2.3241$, respectively. The standard deviation of `lossesin` and `lossesout` are 0.005654 and 0.005535, respectively. We increase the number of buckets as $\{10, 20, \ldots, 100\}$. Finally we report the multiplicative approximation error of our estimates w.r.t. the true distance in both the cases with increasing number of buckets. We run each of the experiments 50 times.

**Results:** Given 1000 samples, Figure 6 show that `STVA` computes the TV distance between output distributions for IN and OUT datasets. The sample streams arrive via $S = 10$ sources. The approximation error drops below 0.1 while using only 250 buckets. This shows `STVA` can conduct resource-efficient privacy auditing of large-scale datasets.

## G.3  Enlarged Plots

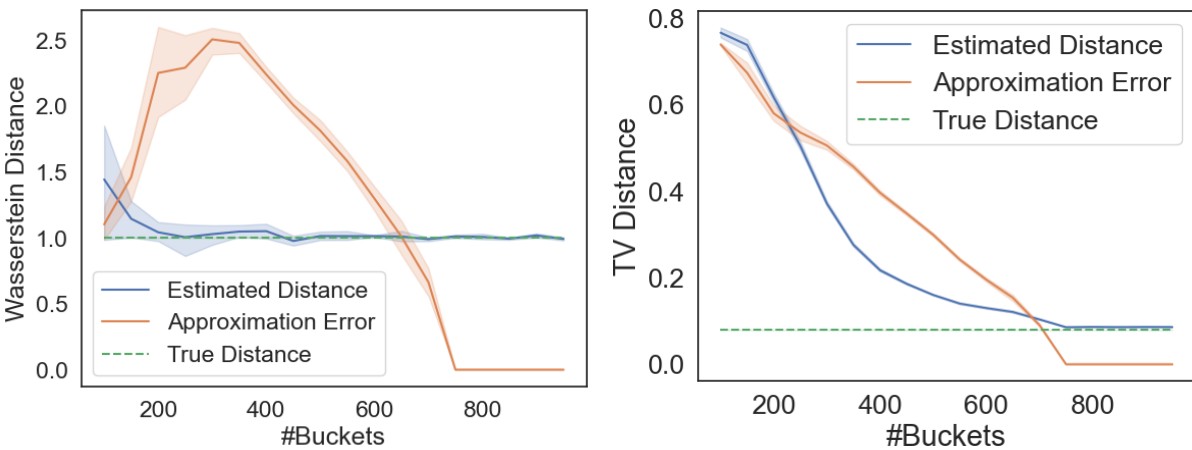

Figure 7: Performance of `SWA` with $\mathcal{N}(0,5)$, $\mathcal{N}(1,5)$ and $b = 0.05$

Figure 8: Performance of `STVA` with $\mathcal{N}(0,5)$, $\mathcal{N}(1,5)$ and $b = 0.05$

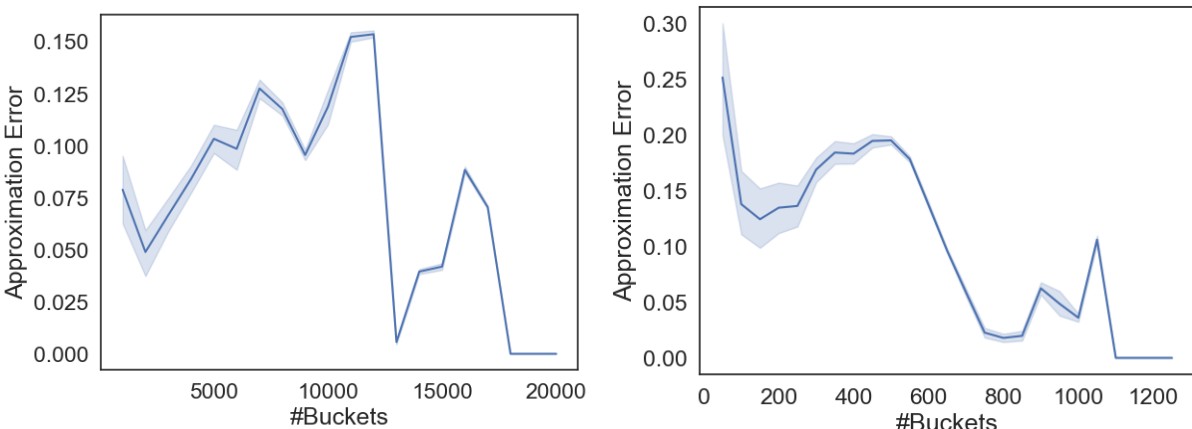

Figure 9: Auditing with `SWA` on regression output of ACS_Income

Figure 10: Auditing with `SWA` on classification output of ACS_Income

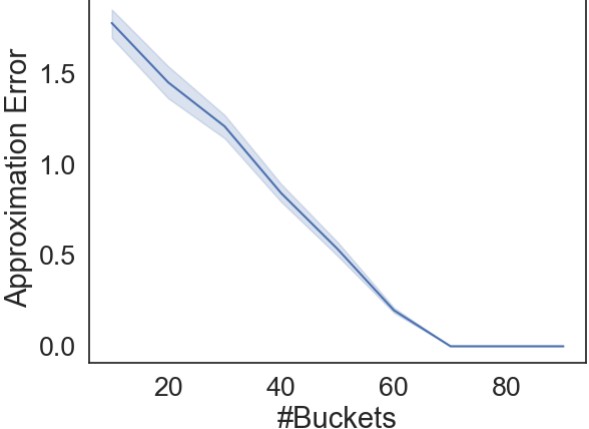

Figure 11: Privacy Auditing of logistic regression on MNIST.

