# OpenReview forum: "Sublinear Algorithms for Estimating Wasserstein and TV Distances: Applications to Fairness and Privacy Auditing"
_TMLR — Accepted by TMLR_

### Review · Reviewer_DJUY · 2025-10-27

**Summary Of Contributions:**

This paper makes a strong theoretical contribution to sublinear distance estimation by unifying discrete and continuous settings and providing explicit error bounds. However, its practical generality remains limited: the framework targets only univariate cases and is evaluated on simple datasets with simple models consisting of logistic and linear regressions. These experiments confirm theoretical soundness but offer limited guidance for practical use. Showing even small-scale or conceptual examples that connect the framework to real-world AI workflows would make the paper substantially more appealing to applied researchers.

**Audience:**

Yes

**Audience Explanation:**

The paper addresses a meaningful and fundamental problem. Efficiently representing probability distributions with reduced computational and memory costs has broad theoretical and practical implications.

**Claims And Evidence:**

Yes

**Claims Explanation:**

Pros:
The proposed framework can be considered more general than prior approaches, at least in a theoretical sense. It unifies discrete and continuous settings, extends the tail assumptions from sub-Gaussian to sub-Weibull distributions, and supports multiple distance metrics under a single mergeable structure.
Finally, the paper provides explicit theoretical error bounds for the proposed mergeable sublinear summaries, which is an important and often overlooked aspect in prior work. Understanding and bounding the approximation error is crucial when summaries are merged, since small local inaccuracies can compound during aggregation. This paper fills that gap by establishing provable error guarantees for both sub-Gaussian and sub-Weibull distributions, thereby strengthening the theoretical foundation of mergeable summary methods.
Cons:
The practical generality of the proposed framework is limited. While the theoretical formulation is elegant, it only applies to univariate distributions over ℝ in practice. Most real-world datasets are inherently multivariate, and the paper does not discuss how the proposed methods could extend to higher-dimensional settings. As a result, some readers may find it difficult to see how this framework could be applied beyond simple one-dimensional cases.

Furthermore, this limitation is reflected in the experimental design. While the theoretical contributions of the paper are well established and do not necessarily require extensive experimentation, the work provides limited illustrative examples that could help AI researchers see how the proposed framework might be applied in practical machine learning contexts. The experimental evaluation focuses only on simple datasets such as ACS Income and MNIST, using basic linear and logistic regression models. These settings are sufficient to validate theoretical correctness but offer little insight into how the framework could inform or enhance real-world ML practice, where datasets are more complex and model architectures far richer.

**Requested Changes:**

While the theoretical side of the paper is already well grounded and does not require extensive experimentation, the work would benefit from showing how the framework could connect to practical ML scenarios. Even a few illustrative hints—such as potential use cases or small-scale examples demonstrating integration with existing AI pipelines—would make the contribution more compelling and accessible to applied researchers.

---

> ### Author Response · Authors · 2025-12-16
> **Authors' response**
>
> We thank the reviewer for their constructive suggestion and appreciation of the novelty of the work. We respond to the comment regarding the application of the proposed algorithms below.
>
> > Our algorithms can be used as a plug-in estimator in any phase of an AI pipeline where the TV and Wasserstein distance of univariate distributions are to be estimated, or an univariate distribution is to be learned in Wasserstein distance. In Section 5.1, and Appendix G, we outline how to leverage our framework for fairness and privacy auditing. Our framework essentially provides an efficient subroutine for these tasks. In particular, as outlined in Appendix G, estimating the TV and Wasserstein distances are important steps for privacy and fairness auditing of ML models, respectively.

---

### Review · Reviewer_3TFq · 2025-11-06

**Summary Of Contributions:**

This work proposes a method for estimating the pdf and cdf of a distribution from a stream of samples with (a) bounded estimation error while (b) ensuring sublinear space and time complexity in the number of samples.
Having such estimates for two distributions $P,Q$ also allows for estimation of their Wasserstein distance and total variation distance with bounded error. These, in turn, have applications in auditing fairness and privacy, respectively.

The proposed method build upon existing methods for "mergeable summaries of histograms" (MMG). These methods essentially keep track of the $k$ most frequent values from a stream. If $k$ is large enough to contain most values observed in the stream, then it is possible to provide a good estimate of the complete histogram of observed values.
Naturally, for finite $k$ this approach breaks down when the stream of data is sampled from a continuous distribution.

The first contribution, estimation of pdfs and cdfs for continuous distributions, largely proceeds as follows:
All samples coming from the distribution are quantized to allow for computation of a histogram (with infinitely many quantization bins).
Instead of materializing the full histogram with infinite space complexity, or a sparse histogram with linear space complexity, an approximation is computed via MMG with $k$ bins.
Assuming expontentially decaying tails on the continuous distribution, and sufficiently large $k$, it is likely that the sparse histogram is estimated with small error. This is because most values from the stream will fall into one of the $k$ buckets. The authors further prove that these small errors in the MMG histogram translate to a small gap between its empirical pdf/cdf and the pdf/cdf of the continuous distribution.

The second contribution, estimation of Wasserstein distances, can be summarized as follows:
It is known that Wasserstein distance can be computed via the inverse cdf of two distributions.
The authors show that, under mild regularity conditions, the previously derived error bounds on empirical cdfs translate to small differences between the true inverse cdf and the empirical quantile function. This is shown to impliy a good approximation of the Wasserstein distance.

For the third contribution, the authors show similarly that the small errors in approximating the pdf lead to small error in apprxoimating total variation distance.

The proposed methods are evaluated in their ability to approximate known total variation / Wasserstein distance for simple distributions (Gaussians) for varying numbers of MMG buckets. They are further evaluated by plugging them in as better estimators of total variation / Wasserstein distance in two experiments from prior work on fairness / privacy auditing, respectively.

### Strengths
* The proposed method is a sound and natural solution to the considered problem and makes creative use of methods developed for a different purpose
* The method has applications in ML fairness and privacy, including federated settings, and is thus a good fit for the venue.
* The level of abstraction in discussion of the overall proof strategy and presentation of key theoretical results is well-chosen
* The manuscript is indredibly well-structured and has a clear narrative. For example, Section 1 uses a parallel structure to discuss TV / Wasserstein distances / mergeable history, which always ends in some clearly defined research question.
* Related work is discussed in sufficient breadth
* Limitations (end of Section 6) and underlying assumptions are discussed to a sufficient degree. The assumptions are, in fact, explicitly highlighted (see bottom of p.5), and the authors make a good argument for why they are reasonable / not overly restrictive

### Weaknesses
I don't see anything that I would consider a major weakness within the TMLR reviewing framework. Nevertheless, here are some points that I would generally consider a weakness,

* The method assumes scalar distributions. This limits its useful in fairness/privacy auditing (e.g., directly auditing model parameters released by DP-SGD or auditing models other than classifiers/regressors).
* The usefulness for privacy auditing of scalar mechanisms also seems a little overstated. The proposed method can, of course, be used for auditing $(\epsilon, 0)$-DP, i.e., total variation distance for scalar mechanisms. But the discussion of (Koskela & Mohammadi (2024)) in Appendix G.2 implies that it could also be user for auditing $(\epsilon, \delta)$-DP. However, looking at their Lemma 11, it appears like the proposed method can only be used to test some sufficient conditions for a bound on privacy profiles, but not to actually evaluate the bound
* The evaluation on Gaussian distributions in Section 5 is somewhat narrow. What about other Sub-Weibull distributions, other number of samples, other parameters for the Gaussians etc.? But since these figures only serve to illustrate results that have already been formally proven, I would not consider this a major weakness.
* Section 4.2 lacks some motivation, which disturbs the narrative flow. See "Requested Changes" below
* Algorithm 1/2 are not self-contained and thus hard to follow. They refer to a larger algorithm from Appendix A, which is never referenced in Section 4. See "Requested Changes" below

**Audience:**

Yes

**Audience Explanation:**

The connection to fairness/privacy makes this work relevant to the trustworthy ML community, and estimation of optimal transport distances is also obviously relevant for various other sub-fields (e.g., generative modelling.)

**Claims And Evidence:**

Yes

**Claims Explanation:**

Yes, all claims around errors / space complexity / runtime complexity are supported via proofs
* that follow a sound and clearly presented strategy
* whose intermediate technical steps are clearly laid out in the Appendix

As mentioned above, the empirical evaluation based on approximating distances between Gaussian distributions for some narrow range of parameters is somewhat weak. But since these experiments serve more of an illustrative function than to support any claims about SOTA-ness, I would not consider this as violating the acceptance criterion.

Similarly (even though it is not an explicit claim), the discussion of auditing for approximate DP in Appendix G.2 is potentially a little misleading. However, it does not contradict any explicit claims made in the paper.

**Requested Changes:**

### Critical
* Update Appendix G.2 to make it clear whether the proposed method can be used for auditing $(\epsilon,\delta)$-DP or only $(\epsilon, 0)$-DP. If I misunderstood something here, feel free to correct me during the discussion period.

### Presentation (non-critical)
* Repeat the motivation for why the empirical distribution is bucketed before applying MMG in Section 4.2.A. So far, it is only mentioned in passing in the contributions list of Section 1.
* Reference Algorithm 5 in Section 4.2 or add an import statement to Algorithms 1/2. Otherwise, it is not clear what Merge(T2) is supposed to refer, what the difference between Merge and MMG*Merge is, etc.
* Potentially use two different words for "buckets of MMG" and "buckets of the quantized histogram". Maybe "buckets" and "quantization bins"?
* Make the first two italicized research questions in Section 1 consistent. It currently looks as if they had different assumptions: "two continuous distributions with infinite support" vs "two distributions" and "if we obtain a stream of n samples" vs "number of samples n".

### Typos etc.
* Typo "WasseSRstein" in Section 1
* Missing number after "We choose the bucket size to be []" in Appendix G.2
* Missing word in "Mergeable Sublinear Representation of Distributions" paragraph of Section 1: "summary of a distribution from the data stream [THAT] can address the [...]"

---

> ### Author Response · Authors · 2025-12-16
> **Authors' response**
>
> We thank the reviewer for appreciating the rigour and applicability of the work, and their detailed suggestions. We have addressed all the presentation issues (highlighted in blue) and typos in the updated draft. Below, we address some of the points raised by the reviewer.
>
> **1. Usefulness for auditing $(\epsilon, \delta)$-DP**
>
> > Koskela \& Mohammadi (2024) show that a histogram-based estimator of hockey-stick divergence can be used for privacy auditing, and more specifically, to create an upper bound on the privacy profile, i.e. a graph illustrating the change of $\delta$ with $\epsilon$ for a specific mechanism.
>
> > In order to do so, Koskela \& Mohammadi (2024) leverage Lemma 11-- if we can estimate each of the PDFs of the IN and OUT distributions with controllable errors w.r.t. TV distance, we can construct an empirical upper bound on the true hockey-stick divergence. Further, as shown in their Theorem 12, if the error in PDF estimations decay with increasing number of samples, the empirical hockey-stick divergence can act as an asymptotically consistent estimate of the true hockey-stick divergence. If we consider our present analysis of SPA and STVA, we observe that each of the PDFs constructed by SPA are $O(n^{-1/3})$ approximations of the true PDFs, where $n$ is the number of samples.
>
> > Thus, the proposed SPA and STVA satisfies the requirement of an asymptotically consistent estimator of  hockey-stick divergence, and hence, can be used for privacy auditing.
>
> **2: Presentation (Non-Critical)**
>
> > In our framework, 'buckets' of the quantized histogram directly corresponds to 'buckets' of the MMG algorithm. Usually, the 'buckets' of the MMG algorithm is referred to as 'counters'. We have maintained this naming convention when discussing the MMG algorithm itself. However, we feel using the 'buckets' name for the case when the MMG algorithm is operating on an underlying quantized histogram facilitates presentation.

---

> > ### Comment · Reviewer_3TFq · 2025-12-18
> >
> > Thank you for the rebuttal. This addresses most of the issues I raised in my initial review (especially the method's usefulness for privacy auditing).
> >
> > I was somewhat concerned about the formal mistakes pointed out by reviewer 1. But it appears those could be resolved.
> >
> > I therefore think that the paper continues to fulfill both acceptance criteria.

---

### Review · Reviewer_pZMk · 2025-11-20

**Summary Of Contributions:**

The paper studies the problem of estimating the distance, in either the TV or Wasserstein sense, between two distributions. The distributions are assumed to be either sub-Gaussian or sub-Weibull, and have densities satisfying certain Lipschitzness properties. The authors build on previous techniques which estimate distributions using bucketed empirical distributions, for appropriately chosen buckets. The authors also analyze the space complexity of their algorithms. Notably, the authors allow the support of the distributions in question to be infinite.

**Additional Comments:**

Edit: Based on the authors response, it seems the concerns raised in my review have succinct fixes that do not have significant downstream impacts on the paper. As such, I have updated my review and will indicate acceptance.

As a minor final point regarding the Lipschitzness issue. In my review I'm referring to Lipschitzness in the sense the authors have defined via definition 8 (over a continuous domain). However, I see the author's point that this will not affect the mechanics of the proof. An updated definition would be nice if it is important for the paper to use the more general notion of Lipschitzness.

**Audience:**

Yes

**Audience Explanation:**

Ostensibly these result would be of interest to the TMLR community, but only with more explanation on what the Lipschitz and bi-Lipschitz conditions imply. Noting that exponential family distributions satisfy these assumptions is great, but is of limited use without explaining what the Lipschitz constants evaluate to. This is something I believe could be easily fixed, and after that I believe these results would be more useful to the broader community.

**Claims And Evidence:**

Yes

**Claims Explanation:**

Unfortunately, there are several issues with the analysis.

- Lemma 29: It is not true that a bounded second moment implies subgaussianity, nor is this what Proposition 2.5.2. of Vershynin 2018 states.

- Theorem 16 claims the discrete distribution is bi-Lipschitz. How can a non-continuous function be Lipschitz? It seems to me the proof is also incorrect as it does not take into account the case where |a-b| is very small. Sometimes people call the property $|f(a)-f(b)| \leq \ell |a-b| + \tau$, for small $\tau$, the "generalized Lipschitzness" property. I believe this property would be satisfied by the bucketed distribution, and may suffice for the authors proof ideas.

- The paper claims that "Any bounded distribution with continuous and compact support" has bi-Lipschitz CDF. Could the authors elaborate on what bounded means? I don't see how bounded support implies bi-Lipschitzness. Even a distribution over the unit interval could ''spike'' arbitrarily quickly at some point. In any case, this statement does not mean much without some comment on what the bi-Lipschitz constant is.

- Related, the authors claim that Gaussians are Lipschitz. I suppose this is true, but this Lipschitz constant could scale poorly in the variance.

- Are Lipschitz/bi-Lipschitz assumptions on the CDF/PDF standard/reasonable assumptions? I am not overly familiar with the literature on distribution estimation, but it seems odd to assume this in addition to sub-Gaussianity.

Minor comments:

- $n^{Res(\tau)}$ notation is confusing. If this is not $n$ raised to the power $Res(\tau)$, why not just use $Res(n,\tau)$?

- What is does $k^* $ represent in Lemma 4? Why can we pick it to be $k/2$? I could not find an explanation of $k^*$.

- For Algorithm 5 in Appendix A, what does ``$T=T\cup {\xi_i} // |T| \leq k^* + 1 \leq k$ after this line'' mean?

**Requested Changes:**

The errors I mentioned previously need to be fixed if this paper is to be accepted. It may be possible that the analysis still works after these errors are fixed, but unfortunately, these errors occur at early and fundamental points of the analysis. Fixing them may require substantial changes to the paper, which worry won't be practical during the review process.

---

> ### Author Response · Authors · 2025-12-16
> **Authors' response**
>
> We thank the reviewer for the detailed review and pointing out an error in the proofs. We outline the corrections, and answer the questions raised by the reviewer below. We have uploaded a rectified draft with the changes highlighted in blue.
>
> **Rectifying Theorem 10 and Implications:**
>
> > We agree that Lemma 29 is not correct, and consequently the result of Theorem 10 does not hold in its stated form. However, we can use the same proof strategy as Theorem 11 to obtain a very similar result. In particular, this results in a similar statement as in Theorem 11 for $\sigma$-Sub-gaussian distribution with $n \geq c\exp{\frac{\tau^2}{\sigma^2}}\log(1/\delta)$.
>
> > This results in the $n \geq c\log(1/\delta)$ condition for sub gaussian distribution to be changed to $n \geq \frac{c}{\varepsilon}\log(1/\delta)$, similar to the case for sub-Weibull distributions. This change occurs in Theorem 13 (b), Theorem 15, Theorem 17, Theorem 22.
>
> > We want to highlight that this does not change the sample requirement of our main results with respect to Wasserstein distance (Theorem 18), and TV distance (Theorem 23). This is due to the fact that the additional multiplicative $1/\varepsilon$ term is subsumed by the sample complexity required for the \emph{concentration of measure.}
>
> **Lipschitzness of Discrete Distributions and Theorem 16:**
> > 1. A discrete function can be Lipschitz. Although Lipschitz continuity is viewed in the context of functions of the form $f:\mathbf{R}\rightarrow\mathbf{R}$, the definition holds for any metrics space. In particular, given two metric spaces $(\mathbf{X},d_x)$ and $(\mathbf{Y},d_y)$, a function $f:\mathbf{X}\rightarrow\mathbf{Y}$ is said to be $\ell$-Lipschitz if:
> $$
> d_y(f(x_1),f(x_2)) \leq \ell d_x(x_1,x_2) ~~ \forall x_1,x_2 \in \mathbf{X}
> $$
> See, for example, [1] for a more formal definition. An example to consider is $f:\mathbf{Z}\rightarrow\mathbf{Z}$ defined as $f(x) = 2x$, where the set of integers $\mathbf{Z}$ is equipped with the standard metric of $d_z(a,b) = |a-b|$. This function is $2$-Lipschitz.
> For some notable examples of the use of Lipschitzness in the discrete domain, see [2-3].
>
> > 2. The issue of $|a-b|$ being arbitrarily small does not occur for the domain $\mathcal{B}$. In particular, we observe that $b \leq |a-b|$ for any $a,b \in \mathcal{B}$ and $a \neq b$. This enables us to establish the result.
>
> **Sufficient conditions for Bi-Lipschitz CDF:**
> > Here, by bounded distribution, we meant that the distribution has a bounded PDF. We have clarified this issue in the updated draft.
>
> **Lipschitzness of the Gaussian PDF:**
> > A standard calculation to obtain the maximum of the first derivative of the Gaussian PDF shows that the corresponding Lipschitz constant grows as $1/\sigma^2$.
>
> **Applicability of the Assumptions:**
> > Lipschitzness of density function is a classical assumption in distribution learning/estimation. It is a standard assumption even for learning/reconstructing functions.
>
> > Bi-Lipschitzness is a special assumption here for estimating Wasserstein distance. This provides a control over inverse CDF estimation (Lemma 32) required for estimating Wasserstein-1 distance. Since any distribution with bounded PDF, and continuous and compact support satisfies bi-Lipschitz condition, this encompasses most of the common data distributions of interest. Whether it is possible to remove bi-Lipschitzness or show it to be necessary can be an interesting future research direction.
>
> *References:*
>
> [1] Searcóid, M. Ó. Metric spaces. London: Springer London. (2007).
>
> [2] Boucheron, Stéphane, Gábor Lugosi, and Pascal Massart, Concentration Inequalities: A Nonasymptotic Theory of Independence (Oxford, 2013; online edn, Oxford Academic, 23 May 2013)
>
> [3] Ledoux, Michel. “The concentration of measure phenomenon.” (2001).

---

> > ### Comment · Reviewer_pZMk · 2025-12-17
> > **Thanks**
> >
> > Thank you for your detailed response and updating the paper. I have updated my review and providing reasoning in the additional comments section.

---

> > > ### Author Response · Authors · 2025-12-17
> > > **Authors' Response**
> > >
> > > We thank the reviewer for acknowledging the changes made to address their concerns and voting for acceptance. We want to thank you again for the suggestions to improve our work.
> > >
> > > We will promptly include a definition of Lipschitzness in its general form in the paper.

---

### Decision · Action_Editor_2W4w · 2026-01-13

**Recommendation:** Accept as is

**Audience:**

Yes

**Audience Explanation:**

The paper addresses some technical subproblems in fairness and privacy auditing, which are critical topics in trustworthy machine learning which is one central topic of TMLR. Efficient estimation of Wasserstein and TV distances has broad implications for auditing.

**Claims And Evidence:**

Yes

**Claims Explanation:**

The paper provides a solid theoretical contribution to estimation of Wasserstein and Total Variation (TV) distances between 1-d distributions, with applications to fairness and privacy auditing via existing methods that rely on such estimates. This is a timely topic due to the importance of auditing methods. The reviewers raised some concerns about practical scope and also some technical errors in the initial submission, however the authors have addressed them during the rebuttal process, and the claims of the final version seem to be correct. While the experiments are limited to simple datasets and models, they confirm the theoretical results.